# Flagged observation analyses as a tool for scoping and communication in integrated ecosystem assessments

**Hiroko Kato Solvang**[1]*, **Per Arneberg**[2]

**1** Marine Mammals Research Group, Institute of Marine Research, Bergen, Norway, **2** Ecosystem Processes Research Group, Institute of Marine Research, Fram Centre, Langnes, Norway

* hiroko.solvang@hi.no

**Data Availability Statement:** All relevant data are within the manuscript and its Supporting Information files.

**Funding:** Per Arneberg received "Sustainable multi-species harvest from the Norwegian Sea and

## Abstract

Working groups for integrated ecosystem assessments are often challenged with understanding and assessing recent change in ecosystems. As a basis for this, the groups typically have at their disposal many time series and will often need to prioritize which ones to follow up for closer analyses and assessment. In this article we provide a procedure termed Flagged Observation analysis that can be applied to all the available time series to help identifying time series that should be prioritized. The statistical procedure first applies a structural time series model including a stochastic trend model to the data to estimate the long-term trend. The model adopts a state space representation, and the trend component is estimated by a Kalman filter algorithm. The algorithm obtains one- or more-years-ahead prediction values using all past information from the data. Thus, depending on the number of years the investigator wants to consider as "the most recent", the expected trend for these years is estimated through the statistical procedure by using only information from the years prior to them. Forecast bands are estimated around the predicted trends for the recent years, and in the final step, an assessment is made on the extent to which observations from the most recent years fall outside these forecast bands. Those that do, may be identified as flagged observations. A procedure is also presented for assessing whether the combined information from all the most recent observations form a pattern that deviates from the predicted trend and thus represents an unexpected tendency that may be flagged. In addition to form the basis for identifying time series that should be prioritized in an integrated ecosystem assessment, flagged observations can provide the basis for communicating with managers and stakeholders about recent ecosystem change. Applications of the framework are illustrated with two worked examples.

## Introduction

Against a background of increasing impact from climate change and other anthropogenic drivers, causing elevated rates of change in marine ecosystems [1–6] leading to patterns of variability beyond the range of the Holocene [7–12], ecosystem-based management (EBM) is

adjacent ecosystems", funded by The Research Council of Norway (pr. nr. 299554).

**Competing interests:** The authors have declared that no competing interest exist.

increasingly identified as a needed framework for management of marine socio-ecological systems [13]. Integrated ecosystem assessments (IEA) have been developed to provide the scientific basis for EBM [14], and numerous groups of scientists working with IEA have been established, such as the regional IEA groups within the International Council for the Exploration of the Sea (ICES, [15]).

Among the core activities of IEA groups are analyses of time series to summarize changes that have occurred in recent decades in ecosystems, and attempts to highlight possible connections between physical, biological, and human ecosystem components [14, 16]. Emphasis is put on keeping an open communication with management and stakeholders [13, 17, 18]. As the groups typically have at their disposal a large number of time series [16, 19], it will often be necessary to prioritize a subset of them for more extensive analyses and communication purposes [20, 21]. Prioritization should preferably be done using a standardized framework applied to all time series. Here we present an approach which is based on analyses of patterns of recent change, where the aim is to identify time series in which the most recent values deviate significantly from an expected trend, possibly indicating unexpected change. This should be of high relevance for IEA groups, as they are often challenged with understanding and interpreting recent change [14].

Our approach is based on estimating trends of time series before assessing whether the most recent observations deviate significantly from these trends. Since temporal changes in ecosystems can take the form of long-term movements as well as short- or mid-term cyclic periods and noise components, different definitions of trends have been used in marine IEAs [16]. In the field of statistical time series analysis, the long-term movements are commonly classified as 'trends', while short- or mid-term cyclic periods are not, due to the different assumptions about the statistical properties. When investigating a trend in time series data, it can therefore be useful to separately identify non-stationary trends and stationary cyclic components. This decomposition is performed by a framework called 'structural time series modelling', which is using a state space representation where the state of each component is estimated by the Kalman filter algorithm [22]. The concept is different from applying an Autoregressive integrated moving average model to adjust with the aim of studying stationary processes from nonstationary time series data [23].

The Kalman filter algorithm can make one- or multistep-ahead predictions in the numerical procedure. The numerical procedure introduced in this paper uses prediction values and forecast uncertainty bands to assess the status of a recent observation, which determines whether the most recent observation follows the prediction or deviates from it [24], thus giving an indication whether change that is unexpected from the predicted trend, is occurring in the time series. We call the significant deviated observations a "Flagged Observation" and the approach "Flagged Observation analysis", hereafter referred to as "FO" and "FO analysis", respectively. The interpretation of a FO is not equivalent to the types of early warning signals that have been proposed with the aim of predicting critical transitions in marine populations or ecosystems [25, 26], nonlinear ecological change [27] or early warning signs based on theoretical framework in social-ecological networks [28]. An important difference is that while the latter frameworks address change caused by specific types of dynamics and often with specific types of outcomes, FO analysis focuses on identifying unexpected patterns of change that is not specific to any type of underlying dynamics or outcomes. Rather, as described above, FO analysis is a practical tool for IAE groups for prioritizing time series for in depth analyses, communication, and other purposes.

In addition to identify unexpected observations for single years, it can also be interesting to explore whether observations from the most recent years together form a pattern where all are consecutively either above or below the predicted trend in a way that is not expected. This is

equivalent to asking whether there is an unexpected tendency for the most recent years, and a framework for this is also presented here.

In this article, we first introduce the numerical procedure and then demonstrate two examples using time series data for, respectively, the Atlantic Multi-decadal Oscillation (AMO) and the Norwegian Sea ecosystem.

## Statistical method

The statistical method includes first a procedure for trend estimation and second a procedure for FO analysis based on multistep ahead prediction values. The methods we apply have been established in statistical time series analysis [22, 29–31]. The output from these analyses can be used to identify single observations (for example years) that deviate significantly from the expected trend, and, with the help of an additional procedure, explore whether there are unexpected tendencies seen across all of the most recent observations. The details are as given below.

### Trend estimation procedure

The observation model of a time series is given by:

$$y(n) = t(n) + u(n), \quad n = 1, \cdots, N, \tag{1}$$

where $t(n)$ is the trend component and $u(n)$ is the residual component at time step $n$, assuming Gaussian white noise. In this article, we introduce a stochastic trend model given by a $d$th-order difference equation model and a method for estimating the trend [29, 30, 32, 33]. The observation we analyze is recorded at equally intervals (annually or monthly), thus, we consider $y(n)$ as discrete time series.

The stochastic differential trend model is defined by the $d$th-order difference equation, which was posed as a smoothing problem by ref. [34]. This model allows for more flexible trends than does the polynomial regression model. The stochastic trend model for a variable is expressed in the following way:

$$\nabla^d t(n) = v(n), \tag{2}$$

where $\nabla$ is defined as a difference operator given by $\nabla t(n) \equiv t(n) - t(n-1)$ and $v(n)$ is assumed to be an identical and independent sequence with $v(n) \sim N(0, \sigma_v^2)$ where $N$ is 'n' of a normal (Gaussian) distribution, the mean is 0 and the variance is $\sigma_v^2$. If $d = 1$,

$\nabla^1 t(n) = v(n)$, that is, $t(n) = t(n-1) + v(n)$, where the trend is known as a random walk model. If $d = 2, \nabla^2 t(n) = v(n)$, that is, $\{t(n) - t(n-1)\} - \{t(n-1) - t(n-2)\} = v(n)$, then, $t(n) - 2t(n-1) - t(n-2) = v(n)$. The notation for the difference operator is varied, e.g. $\nabla$ is defined in [29, 35], but $\Delta$ is used in [30]. Provided that the variance of $v(n)$ is sufficiently small, $t_i(n)$ yields a smooth trend. If the variance is not small, $t(n)$ yields a fluctuated trend. We choose the second order difference stochastic model to estimate the trend in this study.

The model can be represented in the following state space form [22, 29, 30, 32, 33], as

$$
\begin{aligned}
\text{system model}: \quad & z(n) = Fz(n-1) + Gv(n), \\
\text{observation model}: \quad & y(n) = Hz(n) + w(n),
\end{aligned}
\tag{3}
$$

where $z(n)$ is the state vector corresponding to the trend component $t(n)$, $v(n)$ is the system noise vector that is assumed to be a Gaussian white noise with mean 0 and unknown variance $\sigma_v^2$, $F, G$, and $H$ indicate integers, vectors or matrices, and $w(n)$ is observation error that is assumed to be a Gaussian white noise with mean 0 and unknown variance $\sigma_w^2$. We call Formula

(3) a linear-Gaussian state space model. The trend component is modelled by the $d$th-order differential equation model. When $d = 1$ in Eq (2),

$z(n) = [t(n)]$, $F = G = H = 1$, that is, the system model in (3) is given by

$$t(n) = 1 \cdot t(n-1) + 1 \cdot v(n).$$

When $d = 2$ in Eq (2),

$z(n) = \begin{bmatrix} t(n) \\ t(n-1) \end{bmatrix}$, $F = \begin{pmatrix} 2 & -1 \\ 1 & 0 \end{pmatrix}$, $G = \begin{bmatrix} 1 \\ 0 \end{bmatrix}$, and the system model is given by

$\begin{pmatrix} t(n) \\ t(n-1) \end{pmatrix} = \begin{pmatrix} 2 & -1 \\ 1 & 0 \end{pmatrix} \begin{pmatrix} t(n-1) \\ t(n-2) \end{pmatrix} + \begin{pmatrix} 1 \\ 0 \end{pmatrix} v(n)$. The observation error $w(n)$ corre-

sponds to $u(n)$ in Eq (1) in this case. A particularly important problem in state-space modelling is to estimate the state $z(n)$ based on the observations of the time series $y(n)$.

The state setting trend component shall now be considered to be based on the set of observations $Y(j) = \{y(1), y(2), \cdots, y(j)\}$. In particular, for $j<n$, the state estimation problem results in the estimation of the future state based on the present and past observations and is called *prediction*. For $j = n$, the problem is to estimate the current state and is called a *filter*. On the other hand, for $j>n$, the problem is to estimate the past state $z(j)$ based on the observations until the present time and is called *smoothing* [30]. The general approach to these state estimation problems is to obtain the conditional distribution of the state $z(n)$ given the observations $Y(j)$. The state-space model given by (3) is a linear model, and the system and observations noises and the initial state $z(0)$ follow a Gaussian distribution. That is, all of these conditional distributions become Gaussian distributions. Therefore, to solve the state estimation problem for the state space model, it is sufficient to obtain the mean vectors and the variance (covariance matrices) of the conditional distributions [30]. In order to obtain the conditional joint distribution of states $z(1), z(2), \cdots, y(n)$ given the observations $Y(n) = \{y(1), y(2), \cdots, y(n)\}$ by applying the Kalman filter algorithm is known to be a computationally efficient procedure [22, 30, 31, 36]. The conditional mean and the variance (covariance matrix) of the state $z(n)$ are denoted by

$$z(n|j) \equiv \mathrm{E}[z(n)|Y(j)]$$
$$V(n|j) \equiv \mathrm{E}[(z(n) - z(n|j))((z(n) - z(n|j))')] \tag{4}$$

where ' means transpose. The Kalman filter algorithm recursively operates the following *one-step-ahead prediction* ($j = n-1$) and *filtering* ($j = n$) to obtain the joint conditional distribution of the state based on the derivation shown in Appendix C of [30]. This is here given as:

*prediction*:

$$z(n|n-1) = Fz(n-1|n-1)$$
$$V(n|n-1) = FV(n-1|n-1) + GQG' \tag{5}$$

*filtering*:

$$K(n) = V(n|n-1)H'(HV(n|n-1)H' + w(n))^{-1}$$
$$z(n|n) = z(n|n-1) + K(n)(y(n) - Hz(n|n-1))$$
$$V(n|n) = (I - K(n)H)V(n|n-1) \tag{6}$$

Here, $z(n|n-1)$ and $V(n|n-1)$ correspond to the conditional mean and conditional variance of the state respectively, $Q$ includes the variance of the system noise, and $K$ is called the Kalman gain. Setting the initial state $z(1|0)$ as zero and the initial variance $V(1|0)$ of the state as an

arbitrary real number (e.g. 0.1), the Kalman gain is calculated by using the initial variance and observation noise. The filtering value $z(1|1)$ is obtained by using observation $y(1)$ and the calculated gain from the filtering procedure (6). Then the next prediction values $z(2|1)$ and $V(2|1)$ are calculated by using $z(1|1)$ and $V(1|1)$ in the prediction procedure (5). The iterative calculation procedure for the state is continued until $n = N$. Recursive methods based on state space representations are known to be efficient for calculating the likelihood functions in discrete-time Gaussian processes, and the state space model and the Kalman filter therefore provide an efficient method for the computation of the likelihood of the time series models [22, 29, 30]. In our study, the model provided in (1) includes the parameter vector $\theta = (d, \sigma_v^2, \sigma_w^2)$. The log-likelihood function $l(\theta)$ of the model (1) is given by:

$$
\begin{aligned}
l(\theta) &= \sum_{n=1}^{N} \log f(y(n)|Y(n-1), \theta), \\
&= \sum_{n=1}^{N} \log \frac{1}{\sqrt{(2\pi)^2 \det \Sigma(n)}} \exp\left( -\frac{1}{2} \Delta y(n)' \Sigma(n)^{-1} \Delta y(n) \right) \}
\end{aligned}
\tag{7}
$$

where $Y(n-1) = (y(1), y(2), \cdots, y(n-1))$, $\Delta y(n) = y(n) - Hz(n|n-1)$, and $\Sigma(n) = H(n)V(n|n-1)H'(n) + \sigma_w^2$. Recall that $V(n|n-1) = FV(n-1|n-1) + GQG'$ in the one-step-ahead prediction and the $Q$ control the smoothness of the estimated trend. The variance $\sigma_v^2$ of the system noise in $Q$ can be optimized to obtain a better fitted trend to the data by using maximum likelihood within an arbitry variance range. The variance $\sigma_w^2$ of the observation noise can be directly set to the variance of the observation in this study. These variances could also be estimated as one parameter of this model by the numerical optimization procedure, such as the Newton-Raphson method [29, 30] applied with a likelihood function. If it is necessary to compare differential stochastic models with different orders for the trend component, the optimum differential order $d$ is identified by the AIC [30, 32, 37]. Based on the maximum log-likelihood, the number of parameters for $d$ and the variances of system noise and observation noise, AIC = $-2l(\theta) + 2 \times$ number of parameters.

After identifying the optimum trend model, the smoothed trend is estimated by a fixed-interval smoother algorithm [29, 30, 33]:

$$
\begin{aligned}
A(n) &= V(n|n)F'V(n+1|n)^{-1} \\
z(n|N) &= z(n|n) + A(n)(z(n+1|N) - z(n+1|n)) \\
V(n|N) &= V(n|n) + A(n)(V(n+1|N) - V(n+1|n))A(n)'
\end{aligned}
\tag{8}
$$

In this study, we set the differential order as $d = 2$ to obtain a smooth trend for all time series data as introduced in refs. [29, 30, 33]. In addition, we apply grid searching within a range for finding an optimum $Q$ that can control smoothness for the trend.

## FO analysis by multistep-ahead prediction values

The above Kalman filter algorithm provides one-ahead prediction. However, it can be expanded to give a long-term prediction of the state, which is a multistep-ahead ($j$-ahead) prediction for $j = 1, 2, 3, \cdots$ [22, 30]. Let us consider a relevant situation. With the Kalman filter, one-ahead prediction for $z(n+1)$ is obtained by $z(n+1|n)$ and variance $V(n+1|n)$. If the future observation $y(n+1)$ are not observed, the calculation is formally conducted by using the data observed until time point $n$ [22, 30]. This gives $z(n+1|n+1) = z(n+1|n)$ and $V(n+1|n+1) = V(n+1|n)$. Then, two-ahead prediction $z(n+2|n)$ and variance $V(n+2|n)$ are obtained by $z(n+1|n)$ and $V(n+1|n)$, respectively. In general, $j$-ahead prediction based on the observations until time point $n$ is obtained by repeating the prediction step $j$ times. The algorithm used to predict

states $z(n+1), z(n+2), \cdots, z(n+j)$, based on the data $Y(n)$ observed until time point $n$, is expressed as follows:

$$
\begin{aligned}
z(n + i|n) &= Fz(n + i - 1|n), \\
V(n + i|n) &= FV(n + i - 1|n)F' + GQG', \quad i = 1, \cdots, j.
\end{aligned}
\tag{9}
$$

[22, 30]. When $Y(n)$ is observed, the relationship between the state and the time series is expressed by the observation model $y(n) = Hz(n)+w(n)$ in (3). The mean and the variance for the distribution of the prediction $y(n+j)$ are given as $y(n+j|n) \equiv E(y(n+j)|Y(n))$ and $u(n+j|n) \equiv \mathrm{Cov}(y(n+j)|Y(n))$, respectively. Here, $E(\cdot)$ and $\mathrm{Cov}(\cdot)$ are notations for the expectation vector and the variance-covariance matrix, respectively (but in this study only variance as the data are univariate) [30]. Using the observation Eq (3), the mean of $y(n+j)$ is expressed by:

$$
y(n + j|n) = E(Hz(n + j) + w(n + j)|Y(n)) = Hz(n + j|n).
\tag{10}
$$

The variance of $y(n+j)$ is given by:

$$
\begin{aligned}
u(n + j|n) &= \mathrm{Cov}(Hz(n + j) + w(n + j)|Y(n)), \\
&= H\mathrm{Cov}(z(n + j)|Y(n))H' + H\mathrm{Cov}(z(n + j), w(n + j)|Y(n)) \\
&\quad + \mathrm{Cov}(w(n + j), z(n + j)|Y(n))H' + \mathrm{Cov}(w(n + j)|Y(n)), \\
&= HV(n + j|n)H' + R(n + j).
\end{aligned}
\tag{11}
$$

Therefore, the prediction distribution for $y(n+j)$ based on data $Y(n)$ is a normal distribution with mean $y(n+j|n)$ and variance $u(n+j|n)$ or a standard deviation $\sqrt{u(n + j|n)}$ [30]. The forecast bands (FBs) are calculated by Eqs (6) and (7) [30, 38]. Note that we define the values given by the multistep-ahead prediction procedure as 'forecast values' because they are calculated by using the previous data in the algorithm.

## Assessing unexpected tendencies

If all the observations from $n+1$ to $n+j$ consecutively fall either above or below the predicted trend, this is recognized as a potentially unexpected tendency. Here we estimate the probability of an unexpected tendency ($p_{\mathrm{ut}}$) by multiplying the upper or lower probabilities, where the upper probabilities are calculated if all the observations from $n+1$ to $n+j$ time points are located above the predicted trend, and the lower probabilities are calculated if they are located below the predicted trend. Because these probability estimates have to be independent when calculating $p_{\mathrm{ut}}$, they cannot be estimated from the observations directly, which have to be considered dependent in time series data. Therefore, upper/lower probabilities are calculated based on the residuals expressing the difference between the observations and the predicted trend, which can be assumed to follow an identical independent distribution. For the calculation of the upper/lower probability, the residual part is given as
$\hat{r}(n + i) = y(n + i) - \hat{t}(n + i), i = 1, \cdots, j$, where $\hat{t}(n + i)$ is given by $Hz(n+i|n)$ in (10), that is, $\hat{r}(n + i)$ is equivalent to the $\hat{w}(n + i)$ obtained by $y(n + i) - \hat{y}(n + i|n)$. The Gaussian distribution that the residual part follows, has a mean 0 and variance $\hat{\sigma}_w^2$. Therefore, we consider the upper tale probability $P(X \geq \hat{r}(n + i))$ if the observations consecutively locate above the estimated trend and the lower tale probability $P(X \leq \hat{r}(n + i))$ if the observations consecutively locate below the estimated trend, where $X$ is the random variable given by the Gaussian distribution with 0 and variance $\hat{\sigma}_w^2$. The probability is also calculated for each time point, denoted as $P_1, \cdots, P_j$. Finally, $p_{\mathrm{ut}}$ is obtained as $P_1 \times \cdots \times P_j$. When interpreting $p_{\mathrm{ut}}$, it can be compared with a general statistical threshold, such as 0.05. However, because the value of $p_{\mathrm{ut}}$ will depend

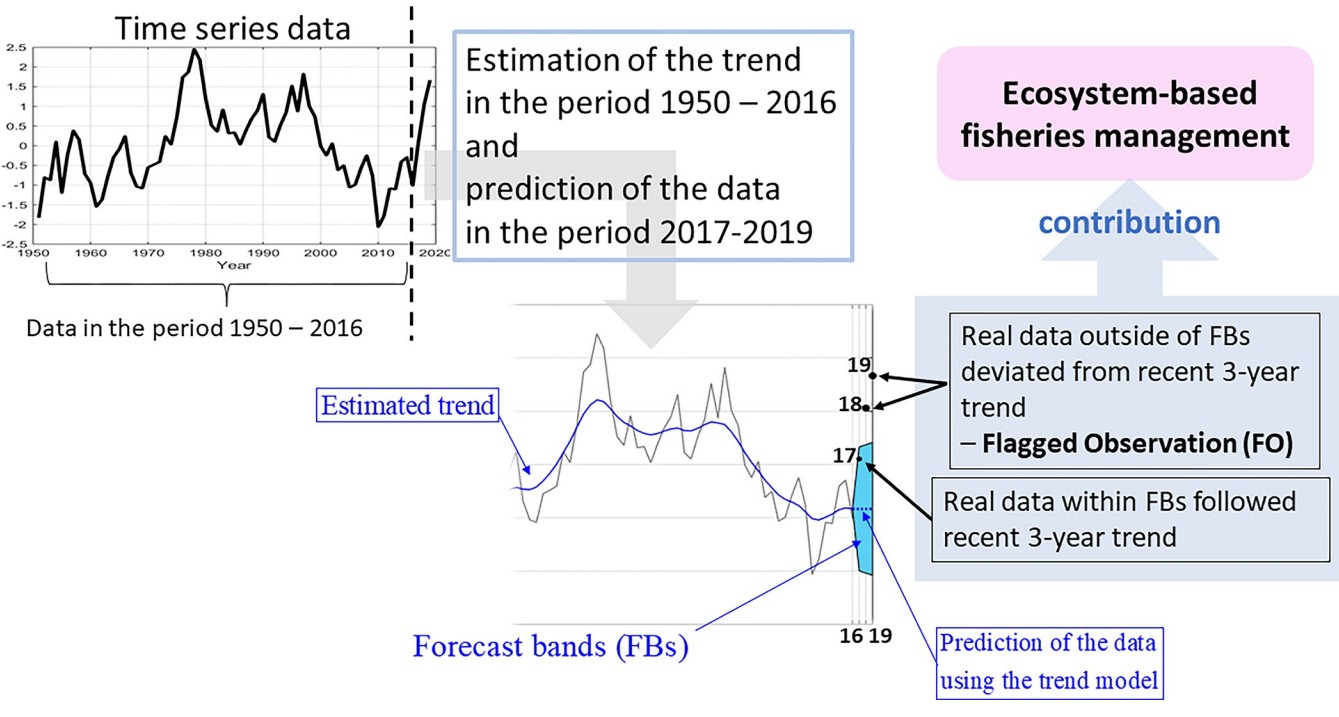

**Fig 1. Outline of proposed flagged observation (FO) analysis.**

on *j*, we recommend that the actual comparison should in this case be made with the threshold $0.05^j$.

The conceptual outlines of this study's analysis procedure for FO analysis and estimation of $p_{ut}$ are given in Figs 1 and 2 respectively. The numerical procedure has been implemented using MATLAB code [39] and R code [40]. MATLAB codes are given in S1 File (grid searching method for Q and the calculation of $p_{ut}$) and S2 and S3 Files (a version using numerical optimization for Q as an option). R code is given in S4 File (grid searching method for Q and the calculation of $p_{ut}$) and S5 File (a version using numerical optimization for Q as an option). Example data are given in S6 File (yearly data of AMO).

## Illustrative examples

The dataset for the Atlantic Multi-decadal Oscillation (AMO) is based on index monthly raw data [41], while for the Norwegian Sea ecosystem, we use the yearly data assembled by the ICES integrated ecosystem assessment working group for the Norwegian Sea (WGINOR, [42]). Abbreviations for the Norwegian Sea data used in this article are summarized in S1 Table together with a short summary of the objective for inclusion of each time series in the work of the group. In the examples, we do not attempt to fully interpret any FOs revealed but comment on the possible background for some of them to illustrate the context for further work in IEA groups. Most of the time series examples in this study are short (i.e., < 50 observations), as this typically what IEA groups have at their disposal [19, 43]. However, some longer time series have also been included.

### The Atlantic Multi-decadal Oscillation (AMO)

AMO is a pronounced signal of climate variability in the North Atlantic Sea surface temperature (SST) [44]. The monthly data recorded from December in 1869 to March in 2021 has

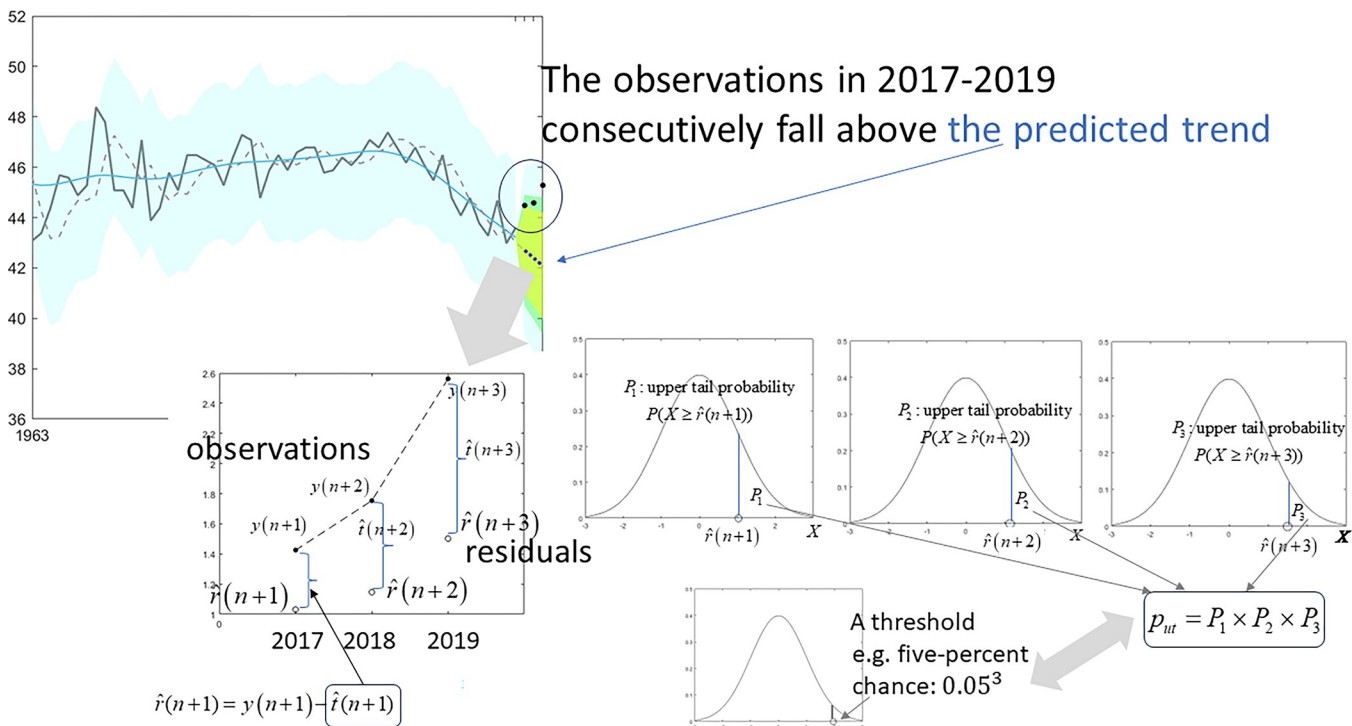

**Fig 2. Outline of the proposed assessment for unexpected tendencies using probability multiplicated by the upper/lower tail probabilities at consecutive time points.**

been published under the NCAR CLIMATE data guide [45]. We extracted monthly raw data for the period 1980–2020, from which we calculated annual means, giving a time series with 41-time points. In addition, a time series with monthly data for the period 1980–2020 has been included to illustrate how seasonality can be taken into account, and a time series for the years 1900–2020 to provide an example of a long time series for these data (results for the latter two time series are shown in the S2 Fig).

To illustrate how inferences may differ for different time periods, both seven-years and three-years predictions are shown for this example. Thus, the three-years ahead prediction is made for the years 2018–2020 and the seven-years ahead prediction for 2014–2020. To optimize $Q$ for the model, we set $0.01 \leq Q \leq 0.1$ as a range of grid serach. The calculated maximum log-likelihood and optimum $Q$ for each dataset are summarized in Table 1. Using the parameters of the model, the Kalman filter algorithm was run to calculate three-years- and seven-years-ahead predictions. Fig 3 presents the outputs of this.

For the case of seven-years-ahead prediction, none of the observations for the most recent years fall outside the 95% FB, but the observation for 2014 fall outside the 70% FB and for 2015 outside the 80% FB. Thus, only two single possible FOs are identified when looking at each of the seven most recent years individually, and this is due to a marked decrease in SST in 2014 and 2015 that contrasts the slightly increasing trend predicted for 2014–2020. Although all observations for the seven most recent years fall below the predicted trend (Fig 3), $p_{ut}$ is not smaller than $0.05^7$ (Table 2), suggesting that there is not a tendency in the data that should be flagged for closer analyses or highlighted in communication with stakeholders. For the case of three-years-ahead predictions, none of the observations from the most recent years fall outside any of the FBs, and they are spread evenly around the predicted trend (Fig 3). Thus, while there are some indications that some of the seven most recent observations deviate from the

**Table 1. Calculated log-likelihood (LL) and optimum Q by applying a stochastic trend model for *d* = 2 for the AMO and Norwegian Sea ecosystem datasets.**

| Dataset | Variable | Q | Maximum log-likelihood |
|---|---|---|---|
| AMO | AMO | 0.01 | -49.9 |
| Norwegian Sea ecosystem (WGINOR) | RHC | 0.01 | -86.9 |
| | RFW | 0.01 | -84.7 |
| | NAO | 0.01 | -170.3 |
| | ZooB | 0.01 | -29.4 |
| | MacB | 0.01 | -41.3 |
| | MacR | 0.01 | -40.1 |
| | MacW | 0.01 | -44.4 |
| | MacL | 0.01 | -73.6 |
| | HerB | 0.01 | -129.3 |
| | HerR | 0.01 | -43.4 |
| | HerW | 0.01 | -88.8 |
| | HerL | 0.01 | -96.6 |
| | BWB | 0.01 | -43.4 |
| | BWR | 0.02 | -50.0 |
| | BWW | 0.01 | -44.6 |
| | BWL | 0.01 | -58.3 |

trend predicted for the last seven years, no such pattern is seen for the trend predicted for the last three years, suggesting that an assessment group may need to look differently at change over these two time periods.

Using a longer time series (1900–2020) to estimate the predicted trend for the last three or seven years, respectively, did not change these conclusions (S2 Fig). Analyses based on monthly data gave predicted trends that follows seasonal fluctuations, and no FOs were detected for the last three or seven months (based on a three-months-ahead and seven months-ahead predicted trend, respectively, S2 Fig).

## Norwegian Sea ecosystem

The Norwegian Sea is located West and Northwest of Norway, bordered by the North Sea and the Atlantic Ocean to the south, the Greenland Sea to the west and the Arctic Ocean and Barents Sea to the north and east. It is a deep-sea area with three species of mainly planktivorous pelagic fish making up the economically most important fish stocks: mackerel (*Scomber scombrus*), Norwegian spring-spawning herring (*Clupea harengus*) and blue whiting (*Micromesistius poutassou*). Ocean currents are dominated by relatively warm and saline Atlantic water masses flowing in from the south and colder and fresher Arctic water masses flowing in from the northwest [46]. There is considerable negative density dependence acting on biomass within the three pelagic fish stocks, presumably through intraspecific competition over food [47]. While there are also indications of competition among the stocks, most strongly between mackerel and herring [47], other work has suggested that interspecific competition is less significant [48]. The combined biomass of the three species has increased over the last decades while zooplankton biomass has declined, and it has been hypothesized that the pelagic fish biomass may have exceeded the carrying capacity of the system [21]. The climate has historically varied between cold and warm phases, with plankton and fish productivity tending to increase in the warmer phases [49]. Here, we analyse time series on spawning stock biomass, recruitment, and growth (age and weight at age 6) for the three pelagic fish stocks, zooplankton

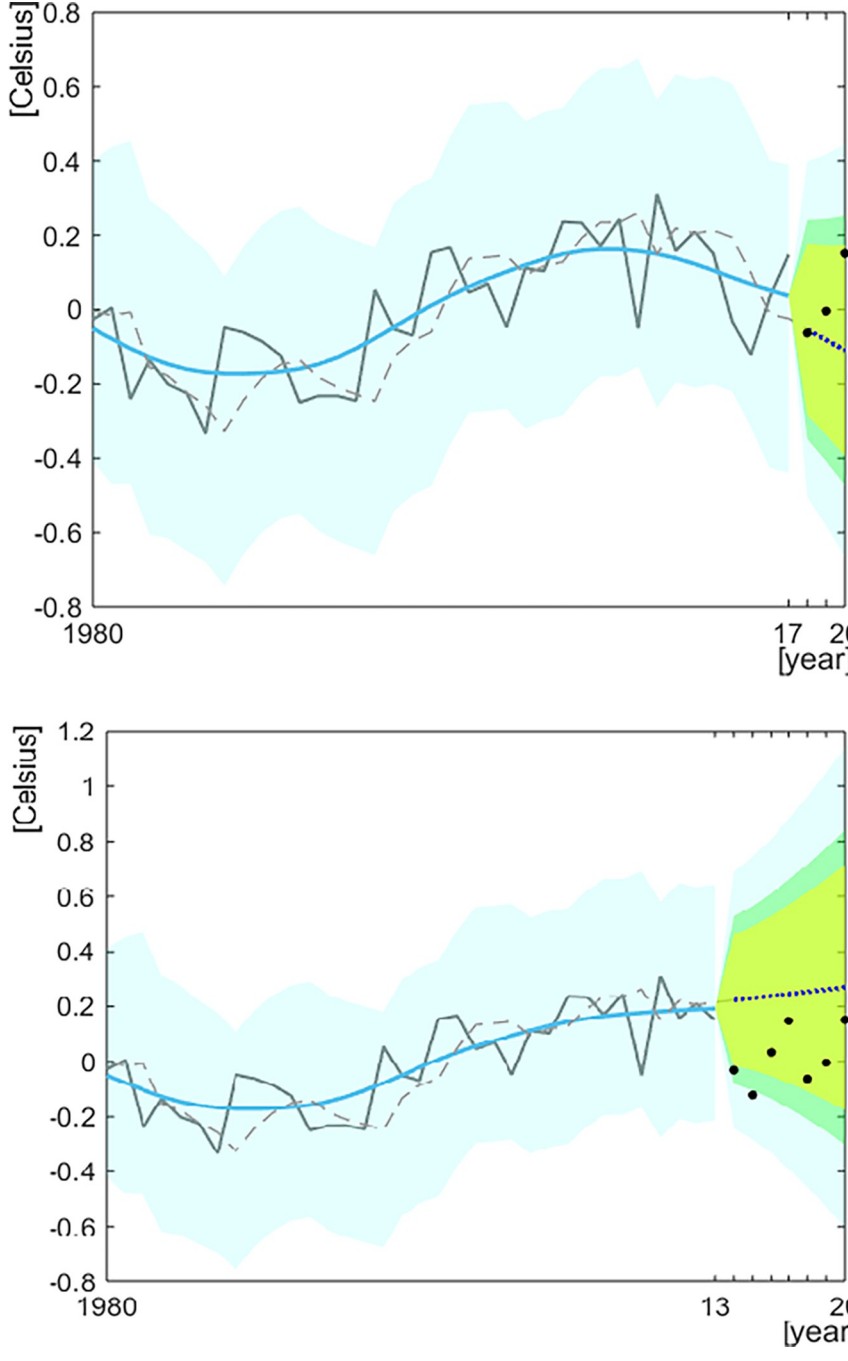

**Fig 3.** The estimated three-years (upper) and seven-years (lower) prediction values of sea surface temperature and the most recent observations for the dataset on the Atlantic Multi-decadal Oscillation. The solid grey lines and the black points indicate the observations used for making the forecast values, and the black points indicate the observations that were plotted for comparison with the prediction values (dotted blue line). The dotted grey line presents the prediction value $Hz(n|n-1)$ and the solid blue lines present the smoothed trend estimates obtained by a fixed-interval smoother algorithm. The light-blue band presents the 95% FB (shown for the whole time series), the dark green band represents the 80% FB and light-green band the approximately 70% FB (the latter two shown for predicted years only).

**Table 2. The upper /lower probabilities at consecutive time points and the $p_{ut}$ for AMO.**

| Year | AMO |
| --- | --- |
| 2014 | 0.062 |
| 2015 | 0.016 |
| 2016 | 0.11 |
| 2017 | 0.28 |
| 2018 | 0.028 |
| 2019 | 0.054 |
| 2020 | 0.24 |
| $p_{ut}$ | 1.1e-08 |

biomass, and three key variables for the physical environment: heat content, freshwater content, and the North Atlantic Oscillation index. The observations have been recorded annually, although the starting/ending years of observation vary among the time series.

For the current work, we used time series with different start years and 2019 as the last year [42]. As one of the main aims of IEAs in the Norwegian Sea has been to provide background information for advisory work for operational fisheries management [50, 51], change over a short period of the most recent years is typically of interest. The conditions for making the prediction were therefore set to three-years-ahead predictions for 2017–2019 using the data observed up to 2016 to estimate the predicted trend. The calculated maximum likelihood, and optimum $Q$ to search in the range for each data are summarized in Table 1. The Q for most time series were estimated as 0.01, which gives a smoothed predicted trend. A fixed estimate of $\sigma_w^2$ was used, which was estimated using the observations until 2016.

Fig 4 presents the outputs of the analyses. Looking at variables for the physical environment, FOs for individual years were observed for relative freshwater content (RFW), where the observation for 2018 fall outside the 80% FB and for 2019 outside the 95% FB. In addition, observations for all the last three years fall well above the predicted trend and $p_{ut}$ is close to $0.05^3$ (Table 3), indicating a possible unexpected tendency in the data. While freshening of the Norwegian Sea had been going on for nearly a decade before 2019 [52], these increases in freshwater content point to a recent intensification of this that might require the attention in IEAs of the Norwegian Sea. For the North Atlantic Oscillation (NAO), all observations for the last three years were lower than the predicted trend, but the $p_{ut}$ for the pattern was substantially higher than $0.05^3$ (Table 3), suggesting that the NAO changes should not be prioritized in an IEA.

For zooplankton biomass (ZooB), the estimate for 2017 fall above the 70% FB. The estimates for 2018 and 2019 also fall above the predicted trend, but $p_{ut}$ is considerably larger than $0.05^3$ (Table 3), indicating that there is no clear unexpected upward tendency in the data.

Looking at variables for pelagic fish stocks, the most substantial evidence for unexpected change is seen for blue whiting. For spawning stock biomass (BWB), one of the three most recent years fall above the 80% FB, another above the 70% FB, and the third also well above the predicted trend (Fig 4). The $p_{ut}$ of this pattern is slightly larger but close to $0.05^3$ (Table 3), suggesting that there may be a tendency for an increase in biomass beyond the expected. At the same time, there are indications of unexpected declines in blue whiting individual growth, shown for weight at age 6 (BWW), where all recent observations fall below the 70% FB (Fig 4) and $p_{ut}$ is substantially smaller than $0.05^3$ (Table 3). For the other measure of growth, length at age 6 (BWL), all the three most recent observations are lower than the predicted trend and one also outside the 80% FB (Fig 4), and with $p_{ut}$ close to $0.05^3$ (Table 3), pointing to a similar unexpected tendency as for weight growth. The changes in biomass and growth may be linked,

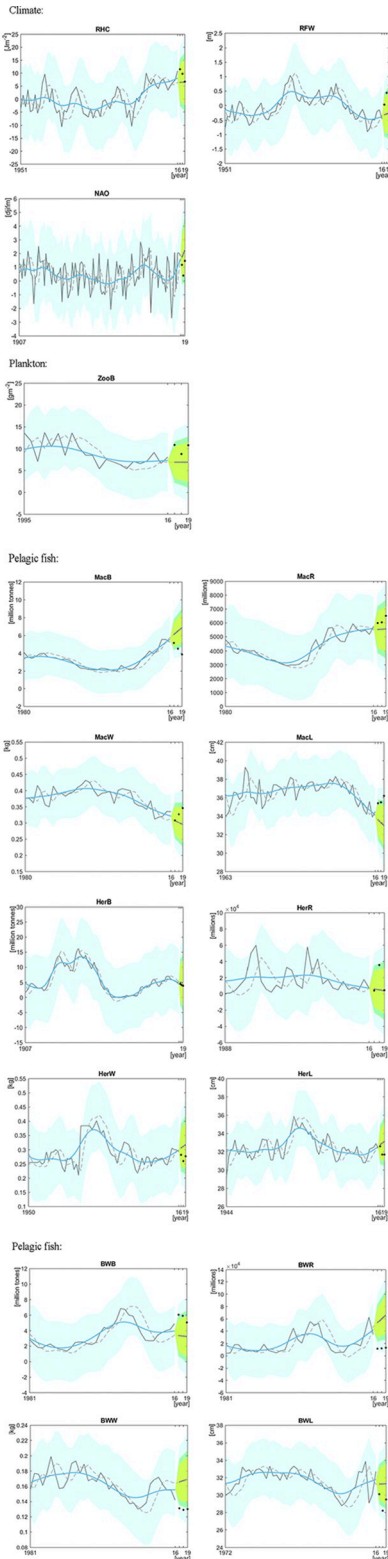

**Fig 4. The estimated three-years prediction values and the three most recent observations for data on variables related to climate, plankton and pelagic fish in the Norwegian Sea ecosystem.** The solid grey lines indicate the observations used for making the forecast values and the black points indicate the observations that were plotted for comparison with the prediction values (dotted blue line). The dotted grey line presents the prediction value $Hz(n|n-1)$

and the solid blue lines present the smoothed trend estimates obtained by a fixed-interval smoother algorithm. The light-blue band presents the 95% FB (shown for the whole time series), the dark green band represents the 80% FB and light-green band the approximately 70% FB (the latter two shown for predicted years only).

as increases in biomass tend to be associated with decreases in growth, possibly through intra-specific competition over food [47]. In addition, for blue whiting recruitment (BWR), two observations fall below the 70% FB and one below the 80% FB (Fig 4) with $p_{ut}$ falling well below 0.05[3] (Table 3), indicating that the decline in recruitment for the most recent years indeed represents an unexpected tendency that should be prioritized in an IEA. Although pelagic fish recruitment remains hard to forecast (e.g. [53]), it is interesting to note that considerable progress has been made in predicting variation in the geographical location of blue whiting spawning habitat, which may be linked to recruitment success [54–56], thus offering a possible avenue for more detailed assessments and studies following up the changes in blue whiting recruitment.

For mackerel, an unexpected decline is seen in spawning stock biomass (MacB), for which two of the three most recent years fall below the 80% FB and the third also below the predicted trend, with $p_{ut}$ well below 0.05[3] (Fig 4, Table 3). As the stock has been fished well above recommended levels since 2013 due to the termination of an international quota sharing agreement [51], a decline in mackerel stock spawning biomass was observed a few years into this period, which is not expected from the predicted trend—indeed an issue that should be flagged for prioritization by an IEA group. Consistent with the decline in mackerel spawning stock biomass, there are indications of an increase above the expected in individual growth measured as length at age 6 (MacL), with observations for one of the three most recent years falling above the 80% FB and the two others above the predicted trend (Fig 4) and with $p_{ut}$ smaller than 0.05[3] (Table 3). While all of the observations for the three most recent years for mackerel recruitment (MacR) fall above the predicted trend, $p_{ut}$ is substantially larger than 0.05[3] (Table 3), suggesting this variable should not be flagged for prioritization in an IEA.

Considering the four variables related to herring, the only FOs for a single year is one estimate of recruitment (HerR), which falls above the 70% FB (Fig 4). As pelagic fish recruitment is highly variable, a single observation falling outside the expected trend may not be a reason for flagging this variable for prioritization in an IEA. Although all observations for herring weight and length at age 6 (HerW and HerL) from the three most recent years fall below the predicted trend, $p_{ut}$ estimates are clearly larger than 0.05[3], suggesting these variables should not be flagged for prioritization.

**Table 3. The upper /lower probabilities at consecutive time points and the $p_{ut}$ for RFW, NAO, ZooB, MacB, MacR, MacL, HerW, HerL, BWB, BWR, and BWW.**

| Year | RFW | NAO | ZooB | MacB | MacR | MacL |
|---|---|---|---|---|---|---|
| 2017 | 0.21 | 0.31 | 0.066 | 0.17 | 0.33 | 0.077 |
| 2018 | 0.047 | 0.075 | 0.24 | 0.026 | 0.32 | 0.040 |
| 2019 | 0.014 | 0.24 | 0.070 | 0.0016 | 0.18 | 0.0052 |
| $p_{ut}$ | 1.4e-04 | 0.0055 | 0.0011 | 7.5e-06 | 0.019 | 1.6e-05 |
| Year | HerW | HerL | BWB | BWR | BWW | BWL |
| 2017 | 0.30 | 0.44 | 0.045 | 0.0041 | 0.033 | 0.20 |
| 2018 | 014 | 0.15 | 0.051 | 0.0013 | 0.020 | 0.0014 |
| 2019 | 0.20 | 0.12 | 0.13 | 0.0004 | 0.018 | 0.099 |
| $p_{ut}$ | 0.0081 | 0.0082 | 2.9e-04 | 2.1e-09 | 1.2e-05 | 2.8e-04 |

## Discussion

We have introduced a time series analysis procedure for making predictions for a specific time period using a structural time series model including a trend model. Based on this, we have outlined a framework for investigating whether the most recent observations deviate from the predicted trend for this time period and thus represent possible flagged observations (FOs). This includes assessing both whether single years represent FOs or whether all the observations from the recent years together represent an unexpected tendency that is classified as a FO. The trend was estimated using a stochastic trend model and observed time series data, and the specific-years-ahead predictions were systematically calculated according to the iterative procedure of the Kalman filter algorithm. The statistical analysis is followed by a qualitative evaluation of each FO, where it may be decided to follow some of them up by more detailed analyses within an integrated ecosystem assessment (IEA). We have also introduced a procedure for assessing the probability that the most recent observations together represent an unexpected tendency ($p_{ut}$). Thus, FO analysis corresponds to a scoping exercise in an IEA that aims at guiding the assessment work and that can also be used for communication purposes. We have also illustrated how FO analysis can be applied using two examples, which illustrate how FOs with different probabilities can be used to guide the work in an IEA. We note that applications of FO analyses may also extend beyond IEAs to other areas of science and advisory processes where an overview of recent change is required across multiple time series.

The time series available from marine ecosystems that are relevant for analyses described here are typically short (i.e., < 50 time points) [19, 43]. This puts constraints on the types of time series analyses that can be performed. For example, null hypothesis testing with time dependency based on autoregressive model or auto-correlation for such short time series can produce misleading results, including false positive and negative results [57]. The procedure described here does not include null hypothesis statistical testing when the trend is estimated, and the type of structural time series model used by us is not based on a frequentist framework but corresponds to a Bayesian approach in the state space representation and Kalman filter algorithm [58], which may properly assess trends in short time series [57]. An alternative method to the one used here could have been the Box Jenkins model, which transforms a non-stationary mean time series to a stationary process [59]. However, effective fitting using this method, again, requires longer time series than what is normally available for marine ecosystems [19, 43]. Thus, for short time series, the Bayesian framework used here appears to be more theoretically appropriate than alternative approaches.

The linear state space model shown in (3) assume a Gaussian distribution for the system noise and observation noise. If a Gaussian distribution cannot be assumed, FOs for single years can still be estimated by applying algorithms such as the extended Kalman filter [31], the non-Gaussian filter and smoothing algorithms by numerical approximating the non-Gaussian distributions by step function, a piecewise linear function, a spline function, and a particle filter [29]. To estimate $p_{ut}$ for non-Gaussian cases, a local regression model [60] or a local linearization filter [61] may be applied.

When assessing the relevance of $p_{ut}$ estimates, we have recommended to use a threshold of, in the case of a 0.05 level, as $0.05^j$. This is done because $p_{ut}$ will depend on the number of recent years considered ($j$). For example, with predictions made for the three last years, this threshold becomes 0.000125 at the 0.05 level. It may be argued that this gives a conservative approach for identifying unexpected tendencies. However, it should be noted that we still identified several unexpected tendencies here (Tables 2 and 3). It is important to note that $p_{ut}$ should not be used in testing for statical significance, but, as for the FOs, as a tool for prioritisation and communication for IEA groups.

To study and assess recent change, IEA groups often rely on examinations of anomaly plots. In such plots, recent change appears as deviations from a long-term mean, often estimated for the whole length of the time series (see e.g. Bulgin, Merchant [62] for an application to global sea surface temperatures, SST). By focusing on deviations from the expected trend for the most recent years (where the expected trend is estimated by using information from the whole time series prior to the recent years), FO analysis can provide a different perspective of recent change. For example, using a seven-years prediction, observations from two of these years fall outside the expected (Fig 3). We argue that the same interpretation is less evident from an anomaly plot of the same time series (see S1 Fig). Thus, while positive and negative values indicate how observations deviate from a *constant* mean value in an anomaly plot, the trend changes through time, making it harder to assess how the most recent observations deviate from the trend that should be expected for these recent years. Recognizing that anomaly plots are important for a large range of purposes within IEAs, we emphasize that FO analysis can provide useful additional information for the practical work in IEA groups, in particular in the light of the challenge they are often faced with of understanding and assessing the most recent development of an ecosystem [14].

Since the cumulative output of FO analysis also aim at giving a sweeping overview of the recent dynamics of all the measured elements in an ecosystem by highlighting the variables that exhibit unexpected change while at the same time showing trends and data for those that do not, the approach should be useful for facilitating the necessary dialogue between scientists and stakeholders about recent ecosystem change within the process of an IEA. Such overviews can also contribute to the scientific output used to educate and inform the public and the political system during parts of policymaking processes related to for example ecosystem-based management [63].

## Supporting information

**S1 Fig. Bar plots for anomalies of AMO's yearly data from 1980 to 2020.** The y-axis indicates the value subtracting mean value of yearly data from 1980 to 2020. The x-axis indicates year. The negative/positive values correspond lower/higher temperature to the mean value. (PDF)

**S2 Fig. Outputs for monthly time series data and longer time series data.** Monthly time series data: The estimated three-months (upper) and seven-months (lower) prediction values of sea surface temperature (blue lines) and the most recent observations for the dataset on the Atlantic Multi-decadal Oscillation from January 1980 to December 2020. The solid grey lines indicate the observations used for making the forecast values and the black points indicate the observations that were plotted for comparison with the prediction values (dotted blue line). The dotted grey line presents the prediction value $Hz(n|n-1)$ and the solid blue lines present the smoothed trend estimates obtained by a fixed-interval smoother algorithm. The light-blue band presents the 95% FB (shown for the whole time series), the dark green band represents the 80% FB and light-green band the approximately 70% FB (the latter two shown for predicted years only).The estimated trends are not smooth as seen for data based on annual means but fluctuate periodically. This is because the monthly data includes seasonal fluctuations; Longer time series data: The estimated three-years (upper) and seven-years (lower) prediction values of sea surface temperature (blue lines) and the most recent observations for the monthly dataset on the Atlantic Multi-decadal Oscillation from 1990 to December 2020. The estimated trend pattern is smoothed, like the trend shown for the shorter time series (1980–2020) in Fig 3, and the FOs are the same as those identified using the shorter time series (Fig 3). (PDF)

**S1 Table. Abbreviations used in figures and tables for data from the ICES integrated ecosystem assessment working group for the Norwegian Sea (WGINOR) and a short summary of the objective for inclusion of each of the time series in the work of the group.** (PDF)

**S1 File. Matlab code for grid searching method for Q and the calculation of $p_{ut}$.** (M)

**S2 File. Matlab code of a version using numerical optimization for Q.** (M)

**S3 File. Matlab code of a function used in S2 File.** (M)

**S4 File. R code for grid searching method for Q and the calculation of $p_{ut}$.** (R)

**S5 File. R code for a version using numerical optimization for Q.** (R)

**S6 File. Yearly data of AMO.** (TXT)

## Acknowledgments

We would like to thank Benjamin Planque, who motivated us to consider this approach for analysing the time series data compiled by the ICES integrated ecosystem assessment working groups and Mette Mauritzen and Daniel Howell for comments on an earlier draft of the paper.

## Author Contributions

**Conceptualization:** Hiroko Kato Solvang, Per Arneberg.

**Data curation:** Per Arneberg.

**Formal analysis:** Hiroko Kato Solvang.

**Funding acquisition:** Per Arneberg.

**Investigation:** Hiroko Kato Solvang, Per Arneberg.

**Methodology:** Hiroko Kato Solvang.

**Software:** Hiroko Kato Solvang.

**Validation:** Hiroko Kato Solvang.

**Writing – original draft:** Hiroko Kato Solvang, Per Arneberg.

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
