## [Decision Letter · Decision Letter 0]

8 Feb 2023

PONE-D-22-30034Flagged observation analyses as a tool for scoping and communication in Integrated Ecosystem AssessmentsPLOS ONE

Dear Dr. Solvang,

Thank you for submitting your manuscript to PLOS ONE. After careful consideration, we feel that it has merit but does not fully meet PLOS ONE’s publication criteria as it currently stands. Therefore, we invite you to submit a revised version of the manuscript that addresses the points raised during the review process.

We look forward to receiving your revised manuscript.

Kind regards,

Pasquale Palumbo

Academic Editor

PLOS ONE

Journal Requirements:

Reviewers' comments:

Reviewer's Responses to Questions

**Comments to the Author**

1. Is the manuscript technically sound, and do the data support the conclusions?

Reviewer #1: Partly

Reviewer #2: Partly

2. Has the statistical analysis been performed appropriately and rigorously? 

Reviewer #1: I Don't Know

Reviewer #2: I Don't Know

3. Have the authors made all data underlying the findings in their manuscript fully available?

Reviewer #1: Yes

Reviewer #2: Yes

4. Is the manuscript presented in an intelligible fashion and written in standard English?

Reviewer #1: Yes

Reviewer #2: Yes

5. Review Comments to the Author

Reviewer #1: I like this paper. To briefly summarize, Kato and Arneberg develop a tool based on the Kalman filter to identify anomalous observations in time series. This tool can be useful in integrated ecosystem assessments where large numbers of observations need to be considered. I do not have full overview of the literature on Integrated Ecosystem Assessments, but Kato And Arneberg provide a satisfying overview of relevant work and they presumably offer a valuable contribution.

The idea is sound and relatively straight forward: Estimate a trend with the Kalman filter and assess whether recent observations deviate sufficiently from this trend to flag them as low probability events. While the basic statistical method is developed and established in the literature and numerous textbooks, the presentation of the technicalities leaves something to desire (see list of comments below).

The approach to assess joint probabilities is non-standard (there is no established standard, as far as I know, unless all involved distributions are Gaussian). This part needs to be explained better. That 'the joint probability ... should be calculated by random variables' (lines 175-177) is cryptic. If I understand correctly, Kato and Arneberg represents the trend prediction uncertainty band by simulation. But if the distribution is assumed Gaussian, I think one can calculate the probability of a given observation from the estimated covariance. Regardless, how the simulated probability evaluation is done in a multivariate setting needs to be spelled out in detail. The relevant computer code is said to be available as supplementary material, but I only found a table of abbrevations in the supplement.

I do not have much to say about the examples, they seem to be executed in a satisfactory fashion. But I wonder why you restrict your sample to the period 1980-2020 (see line 202) when the data series goes back to 1869. And why do you use annual means rather than the monthly observations?

Specific comments:

- Eq. 2: Please provide the definition of the difference operator. The given expression for d=2 is not self-explanatory.

- Eq. 3: What is G? What is its interpretation?

- Eq. 4: What is Q and what is its interpretation? On line 146, you say that Q should be 'optimal'. What is the criteria for the optimal Q? Further, the procedure involving equations 4 and 5 could be explained better. The explanation on lines 123 and forward is unclear.

- Line 137: Why can the variance of w be set to the variance of the observation? Is the variance of the observation an expression of observation uncertainty? Is the observation uncertainty known?

- Line 154: Assuming Y(n+1)=Y(n), where Y(n)=(y(1), ..., y(n)) is cryptic. Presumably, Y(n+1) has the element y(n+1), which evidently is not in Y(n), so it is not clear what is assumed. It is further unclear how z(n+2|n) relates to z(n+1|n).

- Eq. 7: The d-notation is already used for something else (see eq. 2), should be changed.

- Line 203: Annual means for 1980-2020 presumably results in a time series with 41 observations.

Best of luck with further work on this paper!

Reviewer #2: This paper deals with the problem of assessing recent changes in ecosystems by suitably processing time series. The Authors propose a procedure aimed at detecting time series where the most recent observations are less expected on the basis of past data (large deviations from the prediction). The estimation of trends in a time series and the prediction on the basis of past data is obtained by exploiting a stochastic trend model and a Kalman filter/smoother/predictor algorithm. The assessment of unexpected changes is made by using estimated Forecast Bands and joint probabilities. Two examples that illustrate the proposed procedure are reported.

I would like to point out that my opinion on this paper is from the viewpoint of a person with expertise in data processing, and not on biological/ecological applications.

From my viewpoint, the methodology used by the Authors in this paper is well established (it is grounded on the works of Kalman (~1960), Akaike (~1980) and Kitagawa (~1980)). Thus, from a methodolgical point of view I don't see any contribution in this work.

In any case, the Authors should specify in what the proposed "Flagged Observation Analysis" is different from, and possibly an improvement over, the approaches in [25-28].

A possible contribution could be in the two case studies investigated, in the field of marine echosystems, which appear to be interesting, although I am not an expert in this field, so I cannot express a qualified opinion in this area.

Below some comments on data processing aspects of the paper.

After a rather detailed presentation, in the first 6 pages of the paper, of a well established method in the field of time series analysis, the Authors present the proposed methodology for computing "joint probabilities", aimed at "Flagging" unexpected observations and assessing unexpected changes in a time series.

The procedure is outlined in the section "Assessing unexpected tendencies - joint probability for detected FO" in the lines from 180 to 188 of the paper.

However, despite it importance in the paper, the procedure is not clearly introduced and properly justified.

The points that are obscure to me are listed below:

- it is not clear what kind of joint probability is computed in the procedure. It is understood that the computed probability concerns the observations from n+1 to n+J. However, it has not been explained what kind of probability is (is it the probability that ALL the observations from n+1 to n+J are outside a given Forecast Band? Or is the probability that AT LEAST ONE observation in the interval is outside the FB? Or what else?)

- in line 183 I can not understand the sentences "if y(n+j|n) is upper over FB / if y(n+j|n) is lower under FB".

According to my understanding, the predicted value y(n+j|n) is in the middle of any FB, thus how can it be upper or lower w.r.t. FB?

- there is no reported motivation on the chosen sizes of 10.000 (in the generation of random numbers, at step 1) and 1000 (for the averaging in step 3). Is there any quantitative reason for these specific choices?

- since all variables are assumed Gaussian, apparently there is no need to resort to a numerical Monte Carlo procedure, since all probabilities can be accurately computed. Am I missing something?

Throughout the paper the Authors refer to Table 1 and Table 2. However, these tables are not included in the paper.

A suggestion:

I see that the Atlantic Multi-Decadal Oscillation (AMO) are collected monthly, and that the Authors only employ their annual means in their procedure. In this way, the information about the yearly deviation from the mean is lost.

It would be interesting if the authors tested their procedure also on data averaged on a shorter time period (e.g., trimestral or semetral) and compared the obtained results.

Specific comments

-Line 124: specifying the initial variance equal to 0 is rather unusual, except for the particular case where the initial state is exactly known.

- Line 136: The sentence "The flexibility of the estimated trend..." the concept of flexibility should be explained

- In line 141, "c", the argument of AIC(c), should be explained.

- Line 145: if the value d=2 has been chosen exploiting the Akaike criterion (as the Authors say), then the values of AIC should be shown

Typos

- Line 49. Check the sentence starting with "Her we present...", something is wrong.

- Line 121. The Kalman gain in the equations (5) should always written as K(n)

- Line 130. Replace "proceeses" with "processes"

- Line 138. Replace "differernt" with "different"

- Line 146: square brackets should be used for the reference [33]

6. PLOS authors have the option to publish the peer review history of their article (what does this mean?). If published, this will include your full peer review and any attached files.

Reviewer #1: **Yes: **Sturla F. Kvamsdal

Reviewer #2: No

---

## [Author Response · Author response to Decision Letter 0]

1 Jun 2023

Review Comments to the Author

Reviewer #1: I like this paper. To briefly summarize, Kato and Arneberg develop a tool based on the Kalman filter to identify anomalous observations in time series. This tool can be useful in integrated ecosystem assessments where large numbers of observations need to be considered. I do not have full overview of the literature on Integrated Ecosystem Assessments, but Kato And Arneberg provide a satisfying overview of relevant work and they presumably offer a valuable contribution.

The idea is sound and relatively straight forward: Estimate a trend with the Kalman filter and assess whether recent observations deviate sufficiently from this trend to flag them as low probability events. While the basic statistical method is developed and established in the literature and numerous textbooks, the presentation of the technicalities leaves something to desire (see list of comments below).

The approach to assess joint probabilities is non-standard (there is no established standard, as far as I know, unless all involved distributions are Gaussian). This part needs to be explained better. That 'the joint probability ... should be calculated by random variables' (lines 175-177) is cryptic. If I understand correctly, Kato and Arneberg represents the trend prediction uncertainty band by simulation. But if the distribution is assumed Gaussian, I think one can calculate the probability of a given observation from the estimated covariance. Regardless, how the simulated probability evaluation is done in a multivariate setting needs to be spelled out in detail. The relevant computer code is said to be available as supplementary material, but I only found a table of abbrevations in the supplement.

Response: The concept is now described in the introduction. In the methods section, we have now added a description of what we want to estimate: “Thus, we want to estimate the joint probability that the location of the observations from the most recent years relative to the predicted trend could be the result of random variation. As the time series data is sampled as one sample at one time point, this should be calculated by random variables, which is generated from a Gaussian distribution. Thus, what we want to estimate, is the probability that random variation in a Gaussian distribution could have resulted in a lower estimated probability.”

With cross sectional data, the probability of a given observation can be calculated from estimated covariance, which corresponds to a p-value. With time series data, where each observation is from a single time point, this approach cannot be used. Therefore, an alternative approach must be applied, and here we have based this on simulations in the following way: we assume Gaussian distribution for the prediction with Hz(n+j|n) as mean and d(n+j|n) as variance. Furthermore, we conduct Monte Carlo simulations to calculate the p-value. The details are given in the methods section. See lines 185-206.

The computer code has now been added in supplementary files S1 and S2 with example data in supplementary file S3.

I do not have much to say about the examples, they seem to be executed in a satisfactory fashion. But I wonder why you restrict your sample to the period 1980-2020 (see line 202) when the data series goes back to 1869. 

Response: We have mostly focused on short time series (i.e., < 50 observations), because this is typically what is available to IEA groups. In the paper, there are also examples of longer time series (e.g., for herring), and to give additional examples, we have also added analyses using a longer time series for the AMO data and also using monthly data for the AMO time series to provide an example of an even longer time series. These latter analyses have been added to the supplementary material. The main text has been revised accordingly. See lines 218-220, 225-228 and 248-252.

And why do you use annual means rather than the monthly observations?

Response: See above

Specific comments:

- Eq. 2: Please provide the definition of the difference operator. The given expression for d=2 is not self-explanatory.

Response: We have given the definition of the dth-order difference equation in lines 101-107.

- Eq. 3: What is G? What is its interpretation?

Response: G is integer when the trend is random walk model (d=1), and G is a vector leading that the first component corresponds to add v(n). Text describing this is added on lines 113-119.

- Eq. 4: What is Q and what is its interpretation? On line 146, you say that Q should be 'optimal'. What is the criteria for the optimal Q? Further, the procedure involving equations 4 and 5 could be explained better. The explanation on lines 123 and forward is unclear.

Response: We have added the explanation of Q. The criterium for the optimal Q is log-likelihood is as explained in lines143-144 and 153-157. Please note that Q-values had been set too high in the original submission. This has now been adjusted. A modification of the G vector has also been done. This has resulted in some adjustments to the results, most importantly that the decline in mackerel spawning stock biomass is now identified as a flagged observation (lines 317-322).

- Line 137: Why can the variance of w be set to the variance of the observation? Is the variance of the observation an expression of observation uncertainty? Is the observation uncertainty known?

Response: Because the state space model includes the observation model (the second equation) in formulae (3), the variance of the observation is an expression of observation uncertainty. The observation uncertainty is given as the variance of the observations in this study to estimate the variance of the system noise in the system equation (the first equation) in formulae (3), however; the observation uncertainty can be estimated as one parameter of the model by a numerical optimization. We have added an explanation for the flexibility to the observation noise (lines 144-146 and 156-157)

- Line 154: Assuming Y(n+1)=Y(n), where Y(n)=(y(1), ..., y(n)) is cryptic. Presumably, Y(n+1) has the element y(n+1), which evidently is not in Y(n), so it is not clear what is assumed. It is further unclear how z(n+2|n) relates to z(n+1|n).

Response: Since the future observation y(n+1) does not exist in fact, the prediction has to be estimated based on the time series Y(n) = (y(1), y(2), … , y(n)). In the sense, it is assumed that Y(n+1) = Y(n). This concept was referred in Section 9 [57]. We have added the reference in the text (line 164).

- Eq. 7: The d-notation is already used for something else (see eq. 2), should be changed.

Response: We have changed. (lines 177 and 179)

- Line 203: Annual means for 1980-2020 presumably results in a time series with 41 observations. 

Response: text has been corrected (line 225)

Best of luck with further work on this paper!

Thank you very much for your kind words.

Reviewer #2: This paper deals with the problem of assessing recent changes in ecosystems by suitably processing time series. The Authors propose a procedure aimed at detecting time series where the most recent observations are less expected on the basis of past data (large deviations from the prediction). The estimation of trends in a time series and the prediction on the basis of past data is obtained by exploiting a stochastic trend model and a Kalman filter/smoother/predictor algorithm. The assessment of unexpected changes is made by using estimated Forecast Bands and joint probabilities. Two examples that illustrate the proposed procedure are reported.

I would like to point out that my opinion on this paper is from the viewpoint of a person with expertise in data processing, and not on biological/ecological applications.

From my viewpoint, the methodology used by the Authors in this paper is well established (it is grounded on the works of Kalman (~1960), Akaike (~1980) and Kitagawa (~1980)). Thus, from a methodolgical point of view I don't see any contribution in this work.

In any case, the Authors should specify in what the proposed "Flagged Observation Analysis" is different from, and possibly an improvement over, the approaches in [25-28].

Response: The approaches in [25-28] have different scopes, namely identifying change caused by specific types of dynamics and often with specific types of outcomes. Flagged Observation Analysis aims at identifying unexpected change in a way that is not specific to any type of dynamics and/or outcomes, and where the aim is to identify time series that may need to be followed up more closely in an IEA, possibly by addressing dynamics and consequences. We have added text explaining this (lines 75-78).

A possible contribution could be in the two case studies investigated, in the field of marine ecosystems, which appear to be interesting, although I am not an expert in this field, so I cannot express a qualified opinion in this area.

Response: Concerning the contribution of the paper, we would like to point out that the procedure for joint probability calculation is novel. In addition, the application of the methods in an IEA context in the way it is done here is also novel and should be of interest for a wide readership interested in IEAs and related frameworks.

Below some comments on data processing aspects of the paper.

After a rather detailed presentation, in the first 6 pages of the paper, of a well established method in the field of time series analysis, the Authors present the proposed methodology for computing "joint probabilities", aimed at "Flagging" unexpected observations and assessing unexpected changes in a time series.

The procedure is outlined in the section "Assessing unexpected tendencies - joint probability for detected FO" in the lines from 180 to 188 of the paper.

However, despite it importance in the paper, the procedure is not clearly introduced and properly justified.

Response: The need for this procedure has now been introduced and justified in the introduction. Additional description has been added to the methods section (see response to reviewer 1 above).

The points that are obscure to me are listed below:

- it is not clear what kind of joint probability is computed in the procedure. It is understood that the computed probability concerns the observations from n+1 to n+J. However, it has not been explained what kind of probability is (is it the probability that ALL the observations from n+1 to n+J are outside a given Forecast Band? Or is the probability that AT LEAST ONE observation in the interval is outside the FB? Or what else?)

Response: The joint probability is estimated when all observations from the recent years fall on one side of the predicted trend. The position relative to the Forecast Bands is not relevant but was erroneously described as this in the original submission. We have now corrected this and also give a more detailed explanation in the method section. Reviewer 1 raised the same issue, so please see response above. 

- in line 183 I can not understand the sentences "if y(n+j|n) is upper over FB / if y(n+j|n) is lower under FB". According to my understanding, the predicted value y(n+j|n) is in the middle of any FB, thus how can it be upper or lower w.r.t. FB?

Response: As described above, positions relative to FBs are not relevant, but position relative to the predicted trend, and text has been corrected (line 199) .

- there is no reported motivation on the chosen sizes of 10.000 (in the generation of random numbers, at step 1) and 1000 (for the averaging in step 3). Is there any quantitative reason for these specific choices?

Response: Since the state space model assumes a Gaussian distribution, we use a large number as 10000 to generate the distribution to avoid bias in in the estimates. Also, to avoid biased estimation for the joint probability using one Gaussian distribution, we take the averaged joint probability. However, as you pointed, it is arbitrary. Therefore, we have not suggested a specific number of randomizations and iterations but left this open for each investigator to decide (lines 205-206).

- since all variables are assumed Gaussian, apparently there is no need to resort to a numerical Monte Carlo procedure, since all probabilities can be accurately computed. Am I missing something?

Response: This issue was also raised by reviewer 1, so please see the response above.

Throughout the paper the Authors refer to Table 1 and Table 2. However, these tables are not included in the paper.

Response: This must have been a technical problem with the original submission. We are sorry about this. The tables have been uploaded with the revised submission.

A suggestion:

I see that the Atlantic Multi-Decadal Oscillation (AMO) are collected monthly, and that the Authors only employ their annual means in their procedure. In this way, the information about the yearly deviation from the mean is lost.

It would be interesting if the authors tested their procedure also on data averaged on a shorter time period (e.g., trimestral or semetral) and compared the obtained results.

Response: We thank for this suggestion and have added analyses using monthly data (lines 225-227 and 249-252). See also response to the same issue from reviewer 1 above.

Specific comments

-Line 124: specifying the initial variance equal to 0 is rather unusual, except for the particular case where the initial state is exactly known.

Response: No, the initial state z(1|0) is zero and the initial variance V(1|0) is an arbitrary real number (e.g. 0.1). We have rephrased the sentence (lines 130-131).

- Line 136: The sentence "The flexibility of the estimated trend..." the concept of flexibility should be explained

Response: We have corrected ‘smoothness’ from ‘flexibility’ (line 143 - 144).

- In line 141, "c", the argument of AIC(c), should be explained.

Response: We have corrected (line 149).

- Line 145: if the value d=2 has been chosen exploiting the Akaike criterion (as the Authors say), then the values of AIC should be shown

Response: Since we want to estimate the trend as smooth as possible, we fixed d = 2 as indicated in reference [33]. Therefore, in this study, it is not necessary to show AIC to compare the trend model for d = 1 and the trend model for d = 2. We have added the explanation (lines 146-149).

Typos

- Line 49. Check the sentence starting with "Her we present...", something is wrong.

Response: Revised by adding “on” (line 50).

- Line 121. The Kalman gain in the equations (5) should always written as K(n)

Response: We have corrected (line 127).

- Line 130. Replace "proceeses" with "processes"

Response: revised (line 137).

- Line 138. Replace "differernt" with "different"

Response: revised (line 146).

- Line 146: square brackets should be used for the reference [33]

Response: revised (line 154).

---

## [Decision Letter · Decision Letter 1]

19 Jun 2023

PONE-D-22-30034R1

Flagged observation analyses as a tool for scoping and communication in Integrated Ecosystem Assessments

PLOS ONE

Dear Dr. Solvang,

Thank you for submitting your manuscript to PLOS ONE. After careful consideration, we have decided that your manuscript does not meet our criteria for publication and must therefore be rejected.

Specifically:

I am sorry that we cannot be more positive on this occasion, but hope that you appreciate the reasons for this decision.

Kind regards,

Pasquale Palumbo

Academic Editor

PLOS ONE

Additional Editor Comments (if provided):

Both Reviewers have clearly stated that the revised manuscript did not improve the original submission. 

Reviewers' comments:

Reviewer's Responses to Questions

**Comments to the Author**

1. If the authors have adequately addressed your comments raised in a previous round of review and you feel that this manuscript is now acceptable for publication, you may indicate that here to bypass the “Comments to the Author” section, enter your conflict of interest statement in the “Confidential to Editor” section, and submit your "Accept" recommendation.

Reviewer #1: (No Response)

Reviewer #2: (No Response)

2. Is the manuscript technically sound, and do the data support the conclusions?

Reviewer #1: Partly

Reviewer #2: No

3. Has the statistical analysis been performed appropriately and rigorously? 

Reviewer #1: I Don't Know

Reviewer #2: No

4. Have the authors made all data underlying the findings in their manuscript fully available?

Reviewer #1: Yes

Reviewer #2: Yes

5. Is the manuscript presented in an intelligible fashion and written in standard English?

Reviewer #1: No

Reviewer #2: Yes

6. Review Comments to the Author

Reviewer #1: Review, PONE-D-22-30034_R1

The authors have revised their manuscript. As I wrote in my previous report, I think the study is worthwhile documenting in a publication. Reviewer #2 points out that the methodology is known, but I think the application is novel and valuable. Consideration of the behavior of time series needs support by statistical tools. It now occurs to me, however, that the reliance on the normal distribution may be problematic. That the authors do so in their examples may be ok, but I recommend them to add a discussion of issues such as fat tail risk that should be accounted for in real applications. The involvement of non-Gaussian distributions would further highlight the value of the simulation framework they promote, which, as I try to argue below, is unnecessary when working with Gaussian distributions.

My main concern in my previous report was that the presentation of the method needed several clarifications. The authors have tried, presumably, but there are still points that remain unclear (see below). Fortunately, the authors have now supplied their computer code, but I have not had the opportunity to study it. Maybe some of my questions could have been resolved there, but I nevertheless think the manuscript needs more work. Another concern is that the authors claims to have revised the manuscript on specific points, but they have not. In particular, I requested, related to eq. 2, the definition of the difference operator, which I admittedly likely can look up in a statistics textbook. The authors reply that the definition is given in the revised manuscript, but it is not! Further, related to eq. 3, I asked what ‘G’ is and about its interpretation. The authors provide some presumed interpretation in their letter, claiming it is added to the manuscript. One word has been added to the manuscript (‘vectors’), which supplies nothing of the sort. Thus, one may question their motive for engaging in peer-review publication.

My first comment regarded to method for calculating joint probabilities, and I suggested that if distributions are Gaussian, probabilities can be estimated. If I remember correctly, the joint distribution of Gaussian distributions is also Gaussian. The authors claim that with time series, 'where each observation is from a single time point', whatever that is supposed to mean (can an observation be from non-single time points?), the joint distribution stuff does not work. I am not sure, I think considering the joint distribution for consecutive instances of time series are regularly considered in time series statistics (the other reviewer makes the same point), but I am no expert, and I am not sure I contribute to clarifying the situation. Notwithstanding, the authors say they assume a Gaussian distribution for predictions, with some presumably well-defined mean and variance and use Monte Carlo simulations to calculate the p(robability)-value. In their manuscript, the authors write that ‘what we want to estimate with the joint probability here, is the probability that random variation in a Gaussian distribution at each time point, could have resulted in a lower estimated joint probability’ (lines 191-193). While I find this unclear, I maintain that as long as a well-defined Gaussian distribution is assumed, simulations are not necessary to calculate probabilities. Monte Carlo simulations obviously provide a number that is close enough as long as it is appropriately carried out.

With regard to the explanation of the Monte Carlo simulations, I have two comments: (i) On point 2, ‘upper over FB’ and ‘lower under FB’ remains unclear. I failed to comment specifically on this in my previous report; I just requested clarifications in general; I apologize, but these statements are cryptic. The point is further made by reviewer #2, but the statements remain in the manuscript. (ii) On point 4, the summation should run from ‘b’ equals one to M, I presume.

I asked about ‘Q’ in relation to eq. 4, which should be optimal with regard to, it turns out, the log-likelihood (of what?). The authors write that Q-values had been ‘set too high’ in their original analysis. But it is supposed to be optimal, not ‘set’. This is unsettling.

I pointed out that ‘Y(n+1) = Y(n)’ was inconsistent. I realize it is only a notational issue, but it is confusing and looks wrong. The authors insist on sticking to the notation, presumably on grounds that it is used in some reference. Adding this reference does not add clarity to the confusing notation. I find this unfortunate. My related comment that it was unclear how z(n+2|n) relates to z(n+1|n), the authors have simply chosen to disregard.

My reading of the current manuscript is that it remains unclear on several methodological points. While I acknowledge that state space models and the Kalman filter are complex issues that are hard to explain in clear detail, it just makes it more important to be clear and concise. Worse, the authors seem unwilling to revise their work for clarity and provide an honest account of their revisions.

Reviewer #2: As I wrote in my previous review, my opinion on this paper is from the viewpoint of a person with expertise in data processing, and not on biological/ecological applications.

My opinion is that the revised version of the paper has not improved the clarity in the exposition and justification of the proposed method.

My major criticisms concern the section "Assessing unexpected tendencies - joint probability for detected FO", starting at line 184 of the manuscript.

In this section (lines 187-188) the Authors say that the location of the true observations with respect to the predicted trend "could be" the result of a random variation. My point is that of course such locations are the result of a random variation, otherwise the process would be deterministic! The Authors should better characterize the type of randomness that they have in mind. It is a matter of what type of "random model" the future observations are more likely to be drawn from. It is a classical "Hypothesis Testing" problem in statistics.

It is not clear what are the joint probabilities that are estimated by the algorithm from line 195 to line 204.

The given explanation (lines 191 to 193) is not at all clear.

The Authors shoud give a more formal and rigorous definition of the probabilities p_j that are being estimated by the computations at line 199 (step 2 of the algorithm): what are the events captured by such probabilities? Moreover, what are the FO_{n+j}?

Next, the Authors should also give an exact defintion of the joint probability P(p_1,...,p_J) and justify the reason why the events described by the probabilities p_j are independent.

Without a formal definition it is difficult to understand if the estimation procedure is correct, or at least acceptable.

Another point which is not at all discussed, is about the threshold to be considered on the estimated joint probability: when this estimated probability is to be considered so small to flag the observations?

Moreover, as I pointed out in my previous review, since all variables are assumed Gaussian, apparently there is no need to resort to a numerical Monte Carlo procedure, since all probabilities can be accurately computed.

The answer that the Authors gave to this issue is unclear and not satisfactory. A sequence of observations of a Gaussian process is a Gaussian vector. Since the means and covariances are known, all probabilities can be exactly computed.

As a final remark on this section, in my opinion the right methodological framework for the problem investigated by the authors is the Statistical Hypothesis Testing. Indeed, the problem stated by the Authors consists in quantitatively assessing whether the recent observations are in accordance with the estimated past model (Null Hypothesis) or not (Alternative Hypotesis).

Other issues

Lines 155-157: The authors write: "Furthermore, it is possible to identify the optimum Q and R using a numerical optimization based on the log-likelihood function (6)". Thus, it is not clear if they use or not this identification procedure or not (they only say: "is it possible")

Line 159: The section "FO analysis by multistep-ahead prediction values" is not well written.

I understand that observations are available until time n, and the state prediction for the following times is needed.

However, it is not clear the reason why it is formally assumed that Y(n)=Y(n+1)=...=Y(n+j) when j-ahead prediction has to be computed.

When observations are not available, the Kalman filter considers only the computation of the prediction step, without the correction step, and therefore no "formal" output is needed. It would be wrong to compute the Kalman correction step using past values of the observations.

Indeed, the Authors correctly compute the state predictions using the equations at line 170, without the use of any "fake" observation.

Lines 171-174 can be improved. The Authors do not clearly distinguish between the true observation y(n+j) (unavailable) and its prediction y(n+j|n) (computable). The expectation at line 171 is indeed the prediction of the observation at n+j. The covariance at line 172 should be replaced with the error covariance Cov(y(n+j)-y(n+j|n)|Y(n)).

Typos:

Line 119, Either H is a row vector or a transpose symbol is missing in eq. (3)

Line 204. The upper limit of the summation should be M, and not 1000.

Line 230. Maybe 2013 should be 2014.

7. PLOS authors have the option to publish the peer review history of their article (what does this mean?). If published, this will include your full peer review and any attached files.

Reviewer #1: **Yes: **Sturla F. Kvamsdal

Reviewer #2: No

- - - - -

---

## [Author Response · Author response to Decision Letter 1]

14 Aug 2023

Dear PLOS ONE Straive Editorial Assistant, Bernadith Millamina,

This document contains our response to the comments and concerns provided to us by two reviewers on our submitted manuscript:

Re: PONE-D-22-30034

Solvang and Arneberg: Flagged observation analyses as a tool for scoping and communication in Integrated Ecosystem Assessments

The comments from the two reviewers are given in black – how we responded are given in red. We submit the version of the manuscript showing all tracked changes and the line number we have described are corresponding to the version with tracked changes. 

Reviewer #1: Review, PONE-D-22-30034_R1

The authors have revised their manuscript. As I wrote in my previous report, I think the study is worthwhile documenting in a publication. Reviewer #2 points out that the methodology is known, but I think the application is novel and valuable. Consideration of the behavior of time series needs support by statistical tools. It now occurs to me, however, that the reliance on the normal distribution may be problematic. That the authors do so in their examples may be ok, but I recommend them to add a discussion of issues such as fat tail risk that should be accounted for in real applications. The involvement of non-Gaussian distributions would further highlight the value of the simulation framework they promote, which, as I try to argue below, is unnecessary when working with Gaussian distributions.

While we have constructed the linear state space representation by (3) to describe the proposed model (1) and (2) and the state component is estimated by linear Kalman filter, the state space model is not necessary Gaussian distribution. We have explained about the expansion to non-Gaussian model and filtering algorithms in lines 457-467.

My main concern in my previous report was that the presentation of the method needed several clarifications. The authors have tried, presumably, but there are still points that remain unclear (see below). Fortunately, the authors have now supplied their computer code, but I have not had the opportunity to study it. Maybe some of my questions could have been resolved there, but I nevertheless think the manuscript needs more work. Another concern is that the authors claims to have revised the manuscript on specific points, but they have not. In particular, I requested, related to eq. 2, the definition of the difference operator, which I admittedly likely can look up in a statistics textbook. The authors reply that the definition is given in the revised manuscript, but it is not! 

We have described the definition of the difference operator in lines 111-119. is used as a difference operator . The notation is a general form to express the differenciation of the variabes in statistical time series analysis. We have also added the literatures to confirm the notation, e.g. pages 28, 33-34 in [29] and page 19 in [59]. 

Further, related to eq. 3, I asked what ‘G’ is and about its interpretation. The authors provide some presumed interpretation in their letter, claiming it is added to the manuscript. One word has been added to the manuscript (‘vectors’), which supplies nothing of the sort. Thus, one may question their motive for engaging in peer-review publication.

G is a role to operate how the system noise v(n) contributes to which component in the state vector z(n). If we assume random walk trend model, G becomes just scalar 1. If we assume the trend model with d=2, the state vector becomes (t means transpose) and v(n) contribute to only the first component in this state. In the case, G indicates , that is, in (3) is represented by and the first component of contributes the first component of z(n). The system model is given by . We have added concrete explanation about G in lines 128-135.

My first comment regarded to method for calculating joint probabilities, and I suggested that if distributions are Gaussian, probabilities can be estimated. If I remember correctly, the joint distribution of Gaussian distributions is also Gaussian. The authors claim that with time series, 'where each observation is from a single time point', whatever that is supposed to mean (can an observation be from non-single time points?), the joint distribution stuff does not work. I am not sure, I think considering the joint distribution for consecutive instances of time series are regularly considered in time series statistics (the other reviewer makes the same point), but I am no expert, and I am not sure I contribute to clarifying the situation. 

Time series data is recorded as one sample at one time point. For each time point, it is assumed that the data is modelled by the stochastic process. Therefore, ‘mean’ of time series data corresponds to the ‘trend’ component, which is not same for overall time points. In this study, we apply the trend model in (2), which is time dependent, and it is not appropriate to calculate ‘joint probability’ using each probability based on the trend component. Therefore, in this revision, we use the residual part obtained by subtracting trend from the observation, which can be assumed identical and independent Gaussian distribution. Using the residual value, the upper-tale probability (if observations consecutively appear above the trend) probability is calculated using a computational function of the Gaussian distribution function. If consequently time points are three years, we obtain three upper-tale probabilities, e.g. and the joint probability becomes . We have revised the calculation procedure in lines 228-259.

Notwithstanding, the authors say they assume a Gaussian distribution for predictions, with some presumably well-defined mean and variance and use Monte Carlo simulations to calculate the p(robability)-value. In their manuscript, the authors write that ‘what we want to estimate with the joint probability here, is the probability that random variation in a Gaussian distribution at each time point, could have resulted in a lower estimated joint probability’ (lines 191-193). While I find this unclear, I maintain that as long as a well-defined Gaussian distribution is assumed, simulations are not necessary to calculate probabilities. Monte Carlo simulations obviously provide a number that is close enough as long as it is appropriately carried out.

The previous description was to manually calculate upper- or lower-tale probability using random number generated by a Gaussian distribution function. As you pointed out, it is not necessary to use the procedure and we should use the computational function to directly obtain upper- or lower-tale probability of the Gaussian distribution. Furthermore, it is not related to ‘probability value’ that means ‘significant probability’ to support the null hypothesis for statistical testing. We have revised the description in lines 228-259.

With regard to the explanation of the Monte Carlo simulations, I have two comments: (i) On point 2, ‘upper over FB’ and ‘lower under FB’ remains unclear. I failed to comment specifically on this in my previous report; I just requested clarifications in general; I apologize, but these statements are cryptic. The point is further made by reviewer #2, but the statements remain in the manuscript. 

We have excluded the parts related to point 2 and revised the previous procedure, which is better both statistically and computationally. Please see the revised procedures in lines 228-259 and the outline of the procedure in Fig.1b.

(ii) On point 4, the summation should run from ‘b’ equals one to M, I presume.

We have not used the notation in the revised version.

I asked about ‘Q’ in relation to eq. 4, which should be optimal with regard to, it turns out, the log-likelihood (of what?). The authors write that Q-values had been ‘set too high’ in their original analysis. But it is supposed to be optimal, not ‘set’. This is unsettling.

Q is related to used in of the log-likelihood function (6). in is calculated using Q in the ‘prediction’ procedure (4). The log-likelihood is for the model (1). We have added the explanation in lines 173-180. ‘set too high’ meant that the optimum Q value was found within the range we set higher (0.05 to 0.5), however, we re-set lower range (0.01 to 0.1) to obtain more smoothed trend.

I pointed out that ‘Y(n+1) = Y(n)’ was inconsistent. I realize it is only a notational issue, but it is confusing and looks wrong. The authors insist on sticking to the notation, presumably on grounds that it is used in some reference. Adding this reference does not add clarity to the confusing notation. I find this unfortunate. My related comment that it was unclear how z(n+2|n) relates to z(n+1|n), the authors have simply chosen to disregard.

Since the future observation y(n+1) is not observed, the calculation is formally conducted by assuming approximately Y(n+1)=Y(n), which was taken from [34] in the provided procedure. The literature [22] also described that the state and variance for multi-ahead prediction are estimated by the observation until n (Y(n)). This is not our own idea, but is used as a general assumption for multistep-ahead prediction using the linear Kalman filter. We have added the explanation and references in lines 199 - 227. 

My reading of the current manuscript is that it remains unclear on several methodological points. While I acknowledge that state space models and the Kalman filter are complex issues that are hard to explain in clear detail, it just makes it more important to be clear and concise. Worse, the authors seem unwilling to revise their work for clarity and provide an honest account of their revisions.

We apologize for previous unclear explanation. We have added more details to clarify the state space representation and the Kalman filter algorithm for FO detection lines in 136 - 227.

Reviewer #2: As I wrote in my previous review, my opinion on this paper is from the viewpoint of a person with expertise in data processing, and not on biological/ecological applications.

The statistical methodology we used in this study is not novel. As the application using existing statistical methods, we newly provide the FO analysis, which is a practical approach in the Integrated ecosystem assessment for marine resource. Therefore, this paper is about biological/ecological time series application. 

My opinion is that the revised version of the paper has not improved the clarity in the exposition and justification of the proposed method.

We have revised the paper carefully, especially the previous procedure for assessing unexpected tendencies.

My major criticisms concern the section "Assessing unexpected tendencies - joint probability for detected FO", starting at line 184 of the manuscript.

In this section (lines 187-188) the Authors say that the location of the true observations with respect to the predicted trend "could be" the result of a random variation. My point is that of course such locations are the result of a random variation, otherwise the process would be deterministic! The Authors should better characterize the type of randomness that they have in mind. It is a matter of what type of "random model" the future observations are more likely to be drawn from. It is a classical "Hypothesis Testing" problem in statistics.

It is not clear what are the joint probabilities that are estimated by the algorithm from line 195 to line 204.

We apologize for previous unclear explanation. We have revised the previous procedure for the section in lines 228-259 and have made the outline of the procedure in Fig.1b.

The given explanation (lines 191 to 193) is not at all clear.

We have excluded the explanation.

The Authors shoud give a more formal and rigorous definition of the probabilities p_j that are being estimated by the computations at line 199 (step 2 of the algorithm): what are the events captured by such probabilities? Moreover, what are the FO_{n+j}?

We have revised the description to obtain p_j in lines 228-259. 

Next, the Authors should also give an exact defintion of the joint probability P(p_1,...,p_J) and justify the reason why the events described by the probabilities p_j are independent.

Without a formal definition it is difficult to understand if the estimation procedure is correct, or at least acceptable.

As you pointed out, we didn’t explain the independent condition to calculate the joint probability. When the observations locate above the predicted trends at e.g. three time points, for calculating the probability for each time point, the residual obtained by subtracting trend (time-dependent model) from the observation is assumed the variable that obeys identical and independent (Gaussian) distribution. Using the residual value for each time point, the upper-tale probability of the value is calculated by a computational function using Gaussian distribution function. The upper-tale probabilities are independent.

Another point which is not at all discussed, is about the threshold to be considered on the estimated joint probability: when this estimated probability is to be considered so small to flag the observations?

The revised procedure applies an arbitrary threshold 0.05 which means that the event occurs by 5%-tail probability (found in the tail of the distribution). If j = 3, is assessed if .

Moreover, as I pointed out in my previous review, since all variables are assumed Gaussian, apparently there is no need to resort to a numerical Monte Carlo procedure, since all probabilities can be accurately computed.

The answer that the Authors gave to this issue is unclear and not satisfactory. A sequence of observations of a Gaussian process is a Gaussian vector. Since the means and covariances are known, all probabilities can be exactly computed. 

The previous description was to manually calculate upper or lower tale probability using random number generated by a Gaussian distribution function. As you pointed out, it is not necessary to use the procedure and we should use the computational function to directly obtain upper or lower tale probability of the Gaussian distribution. We have revised the description in lines 228-259.

As a final remark on this section, in my opinion the right methodological framework for the problem investigated by the authors is the Statistical Hypothesis Testing. Indeed, the problem stated by the Authors consists in quantitatively assessing whether the recent observations are in accordance with the estimated past model (Null Hypothesis) or not (Alternative Hypotesis).

As you suggested, the procedure is not related to statistical testing considering null hypothesis. It is just calculation of probability based on the Gaussian distribution function. We have excluded ‘probability value’ that means significant probability to assess the test statistics.

Other issues

Lines 155-157: The authors write: "Furthermore, it is possible to identify the optimum Q and R using a numerical optimization based on the log-likelihood function (6)". Thus, it is not clear if they use or not this identification procedure or not (they only say: "is it possible")

In this paper, the variance of the system noise in Q is found by grid search within a range and the variance of the observation noise in R is given by the observation until the year before the year for multi-ahead prediction. On the other hand, there is another way to identify these variances by the numerical optimization procedure (e.g. Newton-Raphson method introduced in [34]). To avoid misunderstanding, we have modified the sentence in lines 181-183. We have also added the program using the numerical optimization procedure in Supplementary file S2.

Line 159: The section "FO analysis by multistep-ahead prediction values" is not well written.

I understand that observations are available until time n, and the state prediction for the following times is needed.

However, it is not clear the reason why it is formally assumed that Y(n)=Y(n+1)=...=Y(n+j) when j-ahead prediction has to be computed.

When observations are not available, the Kalman filter considers only the computation of the prediction step, without the correction step, and therefore no "formal" output is needed. It would be wrong to compute the Kalman correction step usin

---

## [Decision Letter · Decision Letter 2]

22 Feb 2024

PONE-D-22-30034R2Flagged observation analyses as a tool for scoping and communication in Integrated Ecosystem AssessmentsPLOS ONE

Dear Dr. Solvang,

Thank you for submitting your manuscript to PLOS ONE. After careful consideration, we feel that it has merit but does not fully meet PLOS ONE’s publication criteria as it currently stands. Therefore, we invite you to submit a revised version of the manuscript that addresses the points raised during the review process.

We look forward to receiving your revised manuscript.

Kind regards,

João Zambujal-Oliveira

Academic Editor

PLOS ONE

Journal Requirements:

Additional Editor Comments (if provided):

Reviewers' comments:

Reviewer's Responses to Questions

**Comments to the Author**

1. If the authors have adequately addressed your comments raised in a previous round of review and you feel that this manuscript is now acceptable for publication, you may indicate that here to bypass the “Comments to the Author” section, enter your conflict of interest statement in the “Confidential to Editor” section, and submit your "Accept" recommendation.

Reviewer #1: (No Response)

Reviewer #3: (No Response)

2. Is the manuscript technically sound, and do the data support the conclusions?

Reviewer #1: Partly

Reviewer #3: Yes

3. Has the statistical analysis been performed appropriately and rigorously? 

Reviewer #1: Yes

Reviewer #3: Yes

4. Have the authors made all data underlying the findings in their manuscript fully available?

Reviewer #1: Yes

Reviewer #3: Yes

5. Is the manuscript presented in an intelligible fashion and written in standard English?

Reviewer #1: No

Reviewer #3: Yes

6. Review Comments to the Author

Reviewer #1: The manuscript has been revised heavily, mainly in the methodological part, and is now more aligned with standard expositions of the Kalman filter and related algorithms. I still find think the presentation can be written better and be more accessible given that it is an established method. The paper should be read by a copy editor to improve the overall writing and the odd typo. I will not comment in great detail on this section now, the authors have revised in accordance with my earlier comments and comments from the evidently more qualified other reviewer.

I have one comment on an issue that has been discussed earlier, and one question about the relevant probability threshold.

The earlier discussed issue is the case with 'approximately Y(n+1) = Y(n)'. I still find this a strange way of formulating what the authors are doing. I think what they do is to calculate forecasted values over several time steps conditional on Y(n). That is, the forecast for n+2 is conditional on Y(n), and not on Y(n+1) with approximately Y(n+1)=Y(n). As said in earlier correspondence, this may mainly be a notational issue, but the way the authors states it in the present version is cumbersome and could perhaps be wrongly interpreted. I see that the other reviewer also comments on this issue, and importantly points out that no Kalman correction or update should be made without new observations available. This is the kind of misunderstanding that could occur with the authors' notation.

I also wonder about the probability threshold that is used. For predictions over three time steps, the authors compare with the threshold 0.05^3, which is a very small number. Thus, at least given an approximately correct prediction model, very few observations will be flagged. And maybe this is the whole point, if the prediction model is approximately correct, there should be a small chance for flagging observations (false positives). Nevertheless, some discussion along these lines would be valuable, I think.

Reviewer #3: A thorough research has been carried out, which has applied significance. However, some author's assumptions should be given broader explanations, which are given in the review.

Thank you!

7. PLOS authors have the option to publish the peer review history of their article (what does this mean?). If published, this will include your full peer review and any attached files.

Reviewer #1: **Yes: **Sturla Kvamsdal

Reviewer #3: **Yes: **Lupak Ruslan

D.Sc. (Economics), Professor

Department of Economy

Lviv University of Trade and Economics

---

## [Author Response · Author response to Decision Letter 2]

17 Apr 2024

>Reviewer #1

>The manuscript has been revised heavily, mainly in the methodological part,

>and is now more aligned with standard expositions of the Kalman filter and related

>algorithms. I still find think the presentation can be written better and be more accessible

>given that it is an established method. The paper should be read by a copy editor to

>improve the overall writing and the odd typo. 

We have gone through the manuscript carefully to improve the overall writing.

>I will not comment in great detail on this

>section now, the authors have revised in accordance with my earlier comments and

>comments from the evidently more qualified other reviewer.

>I have one comment on an issue that has been discussed earlier, and one question about the

>relevant probability threshold.

>The earlier discussed issue is the case with 'approximately Y(n+1) = Y(n)'. I still find this a

>strange way of formulating what the authors are doing. I think what they do is to calculate

>forecasted values over several time steps conditional on Y(n). That is, the forecast for n+2

>is conditional on Y(n), and not on Y(n+1) with approximately Y(n+1)=Y(n). As said in

>earlier correspondence, this may mainly be a notational issue, but the way the authors

>states it in the present version is cumbersome and could perhaps be wrongly interpreted. I

>see that the other reviewer also comments on this issue, and importantly points out that no

>Kalman correction or update should be made without new observations available. This is

>the kind of misunderstanding that could occur with the authors' notation.

The expression ‘assuming approximately Y(n+1)≈Y(n)’ may lead to misunderstandings. What we meant is that:

Consider the case that I(j) is as a set of actual observed time points in the time points 1,2,⋯,j, Y(j)≡{y(i)|i∈I(j)}.

For one-ahead prediction, the state z(n) and the variance v(n) are then obtained by the Kalman filter using actual observed data series Y(n)= {y(1),y(2),⋯,y(n)}, that is, z(n+1│n)=Fz(n|n) and v(n+1|n)= Fv(n│n) F^'+GQG'. Here, since the future observation y(n+1) is unavailable, Y(n+1)= {y(1),y(2),⋯,y(n)}=Y(n). 

For long-term prediction, we consider j-ahead prediction of the state, z(n+j), based on Y(n+j), that is, z(n+j│n+j)=Fz(n+j-1|n+j-1) and v(n+j|n+j-1)= Fv(n+j-1│n+j-1) F^'+GQG'. 

Since y(n+1),y(n+2),⋯,y(n+j) are not available after observed y(n), Y(n+j)= {y(1),y(2),⋯,y(n)}=Y(n). 

Therefore, the state and variance for j-ahead prediction are z(n+j│n)=Fz(n+j-1|n) and v(n+j|n)= Fv(n+j-1│n) F^'+GQG'. 

The point is that j-ahead prediction based on the observations until y(n) is performed by repeating the prediction step j times, using the relation that Y(n)= Y(n+1)=⋯=Y(n+j). This was from references [22, 34], and is not our own idea.

To avoid a wrong interpretation, we have rephrased the paragraph by using appropriate sentences and references and not using the notation that could lead to misunderstandings (lines 193 - 199). 

>I also wonder about the probability threshold that is used. For predictions over three time

>steps, the authors compare with the threshold 0.05^3, which is a very small number. Thus,

>at least given an approximately correct prediction model, very few observations will be flagged. And maybe this is >the whole point, if the prediction model is approximately correct, there should be a small chance for flagging >observations (false positives). Nevertheless, some discussion along these lines would be valuable, I think.

This threshold value is used to assess the relevance of the put estimates, that is, the probability that the combined evidence of all recent observations indicates that there is an unexpected tendency across all of the most recent (i.e., predicted) years. Therefore, this threshold is not used to assess observations for single years. We agree that using the general formula for a threshold at the 0.05 level for j predicted years as 0.05j may give conservative estimates of what should be considered unexpected tendencies across all recent years. We have added a paragraph in the discussion and thank the reviewer for pointing out this issue (lines 407 - 413).

>Reviewer #3

>A thorough research has been carried out, which has applied significance.

>However, some author's assumptions should be given broader explanations, which are

>given in the review.

> Ecosystems influence the existing conditions of human life, either improve or worsen it, and accordingly require a >person's constant adaptation to such changes. Thanks to observations, it is possible to achieve the necessary >predictability of ecosystems, reliably forecast and satisfy needs while under its influence. All this confirms the >relevance of the article, and despite its significant advantages, it is appropriate to take into account a number of >recommendations: 

>- the article should indicate the objectives of the observations with the overall goal of determining the scope and >communications in ecosystem assessments and the expected results from such observations; 

A summary of the objectives for inclusion of each of the time series in the work of the ICES integrated ecosystem assessment working group for the Norwegian Sea (WGINOR) has been added to Supplementary Table 1 (see lines 249 - 251in the main text). The objective behind the AMO data should be evident from the main text (i.e., it is described as a pronounced signal of climate variability, line 257 - 258).

>- it is worth citing the advantages and disadvantages of conducting the analysis of observations with labels, in >particular, comparing them with some frequently used statistical methods; 

The meaning of the term «label», as used here, is not clear to us. As this could not be clarified after contact with the editor, we are unfortunately not able to address this comment.

>- it is necessary to develop a set of proposals for the active implementation of the observation analysis tool with >labels, addressing them to international institutions, associations of countries or others. 

Again, as the meaning of the term «label», as used here, is not clear to us, we are unable to also address this comment.

---

## [Decision Letter · Decision Letter 3]

7 May 2024

PONE-D-22-30034R3Flagged observation analyses as a tool for scoping and communication in Integrated Ecosystem AssessmentsPLOS ONE

Dear Dr. Solvang,

Thank you for submitting your manuscript to PLOS ONE. After careful consideration, we feel that it has merit but does not fully meet PLOS ONE’s publication criteria as it currently stands. Therefore, we invite you to submit a revised version of the manuscript that addresses the points raised during the review process.

Please take into account the remaining typos indicated by one of the reviewers.

We look forward to receiving your revised manuscript.

Kind regards,

João Zambujal-Oliveira

Academic Editor

PLOS ONE

Journal Requirements:

Additional Editor Comments:

Reviewers' comments:

Reviewer's Responses to Questions

**Comments to the Author**

1. If the authors have adequately addressed your comments raised in a previous round of review and you feel that this manuscript is now acceptable for publication, you may indicate that here to bypass the “Comments to the Author” section, enter your conflict of interest statement in the “Confidential to Editor” section, and submit your "Accept" recommendation.

Reviewer #1: All comments have been addressed

Reviewer #3: All comments have been addressed

2. Is the manuscript technically sound, and do the data support the conclusions?

Reviewer #1: Yes

Reviewer #3: Yes

3. Has the statistical analysis been performed appropriately and rigorously? 

Reviewer #1: Yes

Reviewer #3: Yes

4. Have the authors made all data underlying the findings in their manuscript fully available?

Reviewer #1: Yes

Reviewer #3: Yes

5. Is the manuscript presented in an intelligible fashion and written in standard English?

Reviewer #1: Yes

Reviewer #3: Yes

6. Review Comments to the Author

Reviewer #1: The manuscript looks pretty good. I suspect a remaining typo: An 'L' appears in some of the math where one may expect dots. See for example lines 136, 146, 147, and further on. May be some autocorrect playing games. It seems to have carried over from earlier versions if the track-changes version is to be believed. If so, I apologize for not spotting this earlier.

Reviewer #3: I thank the authors for considering the recommendation and wish them further scientific growth.

Regarding the concept of "labels", we mean "Flagged"

7. PLOS authors have the option to publish the peer review history of their article (what does this mean?). If published, this will include your full peer review and any attached files.

Reviewer #1: No

Reviewer #3: **Yes: **Ruslan Lupak,

Dr Sc (Doctor of Economic Sciences), Prof.,

Prof. of the Department of Economy

Lviv University of Trade and Economics, Ukraine

ORCID ID: https://orcid.org/0000-0002-1830-1800

Scopus Preview ID: https://www.scopus.com/authid/detail.uri?authorId=57189037710

Web of Science Researcher ID: https://publons.com/researcher/2106107/lupak-ruslan

Google Scholar ID: https://scholar.google.com.ua/citations?hl=uk&user=UliL_9wAAAAJ&scilu

---

## [Author Response · Author response to Decision Letter 3]

13 May 2024

Dear PLOS ONE Academic Editor, João Zambujal-Oliveira,

This document contains our response to the comments and concerns provided to us by two reviewers on our submitted manuscript:

Re: PONE-D-22-30034R2

Solvang and Arneberg: Flagged observation analyses as a tool for scoping and communication in Integrated Ecosystem Assessments

The comments from the two reviewers are shown in black font, while our responses are given in red font. In the manuscript, all changes are shown using the “Track Changes” function of MS Word. 

Reviewer #1

>The manuscript looks pretty good. I suspect a remaining typo: An 'L' appears in some of >the math where one may expect dots. See for example lines 136, 146, 147, and further >on. May be some autocorrect playing games. It seems to have carried over from earlier >versions if the track-changes version is to be believed. If so, I apologize for not spotting >this earlier.

This error was introduced when our word file with the manuscript was converted to a pdf file during the submission process. We have identified this as a problem with an old version of the editor used for equations and will see to that the problem is fixed when resubmitting.

Reviewer #3

>I thank the authors for considering the recommendation and wish them further scientific >growth.

>

>Regarding the concept of "labels", we mean "Flagged"

>- it is worth citing the advantages and disadvantages of conducting the analysis of >observations with (labels) flagged, in particular, comparing them with some frequently >used statistical methods; 

We thank the reviewer for clarifying his comments. One of the most common approaches to address change in time series is by using plots and analyses of anomalies. In line 409-423 and Supplementary Fig. 1, we discuss how FO analyses may give useful additional information to that held by analyses of anomalies. In addition, it has recently been shown that commonly used methods for trend detection based on frequentist approaches may produce erroneous results (see ref 54). We point out that the type of structural time series model used by us is not based on a frequentist framework but corresponds to a Bayesian approach in the state space representation and Kalman filter algorithm, which may properly assess trends in short time series (lines 383-391). We therefore think we have covered the two main advantages over other relevant frequently used statistical methods. We see no disadvantages of FO analyses when compared with the above mentioned frequently used methods, but emphasise that for analyses of anomalies, FO analyses represents a supplement rather than a replacement (lines 419-423).

>- it is necessary to develop a set of proposals for the active implementation of the >observation analysis tool with (labels) flagged, addressing them to international >institutions, associations of countries or others.

FO analysis is primarily developed as a tool for scoping in IEA groups (see lines 50-54, 79-80 and 377-378). Thus, the primary uses of the results are the IEA groups themselves. We also point out that the results from FO analyses can be used in communication with stakeholders and that this can also be relevant for policymaking processes (lines 424-430). However, how this should be done in practice, will, in our view, be context dependent and has to be developed by IEA groups and/or users of their results. While we agree that it is highly interesting to discuss this, we feel it is beyond the scope of this manuscript, as it would mean expanding the text considerably by for example including considerable literature on the science-policy interface.

---

## [Decision Letter · Decision Letter 4]

5 Jun 2024

Flagged observation analyses as a tool for scoping and communication in Integrated Ecosystem Assessments

PONE-D-22-30034R4

Dear Dr. Solvang,

We’re pleased to inform you that your manuscript has been judged scientifically suitable for publication and will be formally accepted for publication once it meets all outstanding technical requirements.

Kind regards,

João Zambujal-Oliveira

Academic Editor

PLOS ONE

Additional Editor Comments (optional):

Reviewers' comments:

Reviewer's Responses to Questions

**Comments to the Author**

1. If the authors have adequately addressed your comments raised in a previous round of review and you feel that this manuscript is now acceptable for publication, you may indicate that here to bypass the “Comments to the Author” section, enter your conflict of interest statement in the “Confidential to Editor” section, and submit your "Accept" recommendation.

Reviewer #1: All comments have been addressed

2. Is the manuscript technically sound, and do the data support the conclusions?

Reviewer #1: Yes

3. Has the statistical analysis been performed appropriately and rigorously? 

Reviewer #1: Yes

4. Have the authors made all data underlying the findings in their manuscript fully available?

Reviewer #1: Yes

5. Is the manuscript presented in an intelligible fashion and written in standard English?

Reviewer #1: Yes

6. Review Comments to the Author

Reviewer #1: (No Response)

7. PLOS authors have the option to publish the peer review history of their article (what does this mean?). If published, this will include your full peer review and any attached files.

Reviewer #1: **Yes: **Sturla F. Kvamsdal

---

## [Editor Report · Acceptance letter]

24 Jun 2024

PONE-D-22-30034R4 

PLOS ONE

Dear Dr. Solvang, 

I'm pleased to inform you that your manuscript has been deemed suitable for publication in PLOS ONE. Congratulations! Your manuscript is now being handed over to our production team.

Kind regards, 

on behalf of

Prof. João Zambujal-Oliveira 

Academic Editor

PLOS ONE